# An evaluation of atmospheric absorption models at millimetre and sub-millimetre wavelengths using airborne observations

Stuart Fox[1], Vinia Mattioli[2], Emma Turner[1,5], Alan Vance[1], Domenico Cimini[3,4], and Donatello Gallucci[3]

[1]Met Office, FitzRoy Road, Exeter, EX1 3PB, UK
[2]EUMETSAT, EUMETSAT Allee 1, 64295 Darmstadt, Germany
[3]National Research Council - Institute of Methodologies for Environmental Analysis (CNR-IMAA), 85100 Potenza, Italy
[4]Center of Excellence Telesensing of Environment and Model Prediction of Severe events (CETEMPS), University of L'Aquila, 67100 L'Aquila, Italy
[5]ECMWF, Shinfield Park, Reading, RG2 9AX, UK

**Correspondence:** Stuart Fox (stuart.fox@metoffice.gov.uk)

**Abstract.** Accurate gas absorption models at millimetre and sub-millimetre wavelengths are required to make best use of observations from instruments on board the next generation of EUMETSAT polar-orbiting weather satellites, including the Ice Cloud Imager (ICI), which measures at frequencies up to 664 GHz. In this study, airborne observations of clear-sky scenes between 89 and 664 GHz are used to perform radiative closure calculations for both upward and downward-looking viewing directions in order to evaluate two state-of-the-art absorption models, both integrated into the Atmospheric Radiative Transfer Simulator (ARTS). Differences of 20 K are seen in some individual comparisons, with the largest discrepancies occurring where the brightness temperature is highly sensitive to the atmospheric water vapour profile. However, these differences are within the expected uncertainty due to the observed water vapour variability, highlighting the importance of understanding the spatial and temporal distribution of water vapour when performing such comparisons. The errors can be significantly reduced by averaging across multiple flights, which reduces the impact of uncertainties in individual atmospheric profiles. For upward-looking views, which have the greatest sensitivity to the absorption model, the mean differences between observed and simulated brightness temperatures are generally close to, or within, the estimated spectroscopic uncertainty. For downward-looking views, which more closely match the satellite viewing geometry, the mean differences were generally less than 1.5 K, with the exception of window channels at 89 and 157 GHz, which are significantly influenced by surface properties. These results suggest that both of the absorption models considered are sufficiently accurate for use with ICI.

# 1 Introduction

The Metop-SG series of satellites will be launched from 2025 under the EUMETSAT Polar System - Second Generation (EPS-SG) programme. This new generation of polar-orbiting satellites will provide continuous meteorological observations over the coming decades, and contribute to the Joint Polar System (JPS), a collaborative effort established by EUMETSAT and NOAA. They will carry a suite of new instruments which include, for the first time on an operational mission, a passive sub-millimetre radiometer known as the Ice Cloud Imager (ICI) as well as a more traditional microwave imager (MWI) and sounder (MWS) (Accadia et al., 2020; Mattioli et al., 2019a; Kangas et al., 2012). The primary purpose of ICI is to provide daily global observations of ice cloud properties including total ice mass, ice particle size and ice cloud height (Buehler et al., 2007; Eriksson et al., 2020). ICI has 11 channels measuring between 183 and 664 GHz which complement the lower-frequency channels on the microwave radiometers. Accurate atmospheric gas absorption models are needed in this frequency range to maximise the quality of cloud information that can be derived from ICI measurements (Mattioli et al., 2019b). For thick ice clouds, which are dominated by scattering, the brightness temperatures for cloudy scenes are generally reduced compared to the equivalent clear-sky value, and it is these brightness temperature depressions that will be used as input to cloud retrieval algorithms. Gas absorption models are needed to estimate the clear-sky brightness temperatures from background temperature and humidity fields using radiative transfer models. An error in the absorption model will impact the ability of ICI to detect thin ice clouds, and will also affect the quality of retrieved humidity information (Mattioli et al., 2019a). In addition, comparison of observed and simulated brightness temperatures in clear sky conditions will be used to determine the radiometric accuracy of ICI during post-launch calibration and validation activities, which requires accurate models of atmospheric absorption.

Atmospheric absorption in the microwave and sub-millimetre regions of the electromagnetic spectrum is dominated by water vapour and oxygen, with additional contributions from ozone and nitrogen. The physical basis for absorption from these gases is described by Rosenkranz (1993). Figure 1 shows the contributions of each absorbing species to the optical depth for two atmospheric profiles. These correspond to a typical tropical profile, and a significantly drier profile for comparison.

A major feature of the absorption is the presence of resonant absorption lines for water vapour, oxygen and ozone, which are associated with transitions between different molecular energy states. These lines can be described by a universal shape function, along with a set of parameters which determine their frequency, strength and width as functions of temperature and pressure. Individual line parameters come from different sources including theoretical calculations and laboratory or field measurements. Line databases such as HITRAN (Gordon et al., 2022), the AER line database (Cady-Pereira et al., 2020), GEISA (Jacquinet-Husson et al., 2016) and JPL (Pickett et al., 1998) collect the parameters for many different lines and are updated in response to new studies. However, simply summing the contributions from the individual lines in these databases is insufficient to calculate the total absorption at microwave and sub-millimetre wavelengths. Additional effects need to be accounted for, including line mixing and the dry and water vapour continua.

Line mixing affects the oxygen absorption due to the large number of closely spaced transitions, and acts to modify the line shape (Rosenkranz, 1975), often far beyond the proximity of the lines affected. The biggest impact is in the 50-60 GHz band where there are many closely spaced lines. However, it also has a non-negligible effect on the 118.75 GHz line and in the

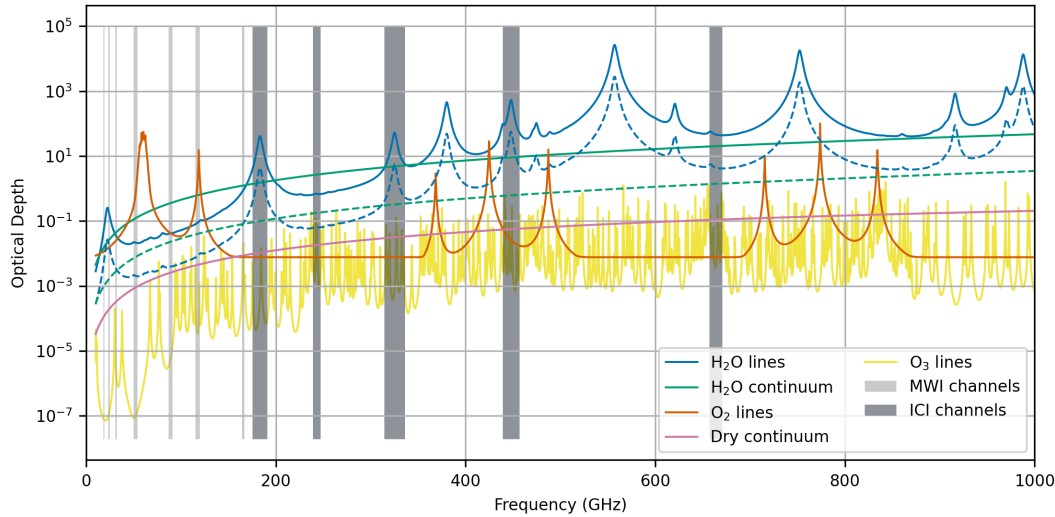

**Figure 1.** Optical depth due to different species for a typical tropical atmosphere (integrated water vapour (IWV)≈56 kgm$^{-2}$). The dashed lines correspond to a profile with IWV≈5.6 kgm$^{-2}$. The gray shading indicates the frequency bands covered by MWI and ICI. Note that both MWI and ICI have channels centred on the 183.31 GHz water vapour absorption line.

89 GHz window region. The dry continuum is a result of collision-induced absorption between pairs of nitrogen and oxygen
molecules. The source of the water vapour continuum is still under debate (Serov et al., 2017; Shine et al., 2012) and there is no complete physically-based description. It is likely to be influenced by collision-induced absorption, water vapour dimers (Serov et al., 2014; Tretyakov et al., 2013), and the contributions of the far-wings of spectral lines (Clough et al., 1989; Serov et al., 2017), but the relative importance of these contributions is uncertain.

A number of "complete absorption models" are available which can be used to calculate total absorption by atmospheric
gases. Here, the term "complete absorption model" indicates that both line and continuum absorption are included in a consistent way. Examples include versions of the Millimetre-wave Propagation Model (MPM) (Liebe, 1989; Liebe et al., 1993) and the series of models developed by Rosenkranz (2017). For computational efficiency, these models typically combine a reduced set of key absorption lines along with appropriate continuum parametrizations. The LBLRTM and MonoRTM line-by-line radiative transfer models developed by AER inc. (Clough et al., 2005) are supplied with a dedicated line database (Cady-Pereira
et al., 2020) and the associated MT-CKD continuum model (Mlawer et al., 2012). A reduced set of lines is also provided to permit faster calculations in the microwave region whilst still retaining sufficient accuracy. The RTTOV fast radiative transfer model (Saunders et al., 2018) is widely used for satellite retrievals and data assimilation. It uses a fast parametrization of atmospheric transmittances and is trained using output from a line-by-line absorption model. At microwave frequencies the AMSUTRAN model (Turner et al., 2019) is used. RTTOV will be used by the operational ICI retrieval algorithms (Eriksson
et al., 2020) and for direct assimilation of ICI radiances in NWP models. The absorption model within AMSUTRAN has therefore recently undergone development to improve its representation in the sub-millimetre frequency range.

Given the diverse sources for spectroscopic parameters and the challenging nature of laboratory experiments it is necessary to validate absorption models under representative atmospheric conditions. This is often done using radiative closure experiments where radiative transfer models are used to simulate brightness temperatures for known atmospheric profiles. The simulated brightness temperatures are compared to observations from ground-based, airborne or satellite radiometers. Spectroscopic parameters may be adjusted as a result of these experiments to improve the agreement between observed and simulated brightness temperatures. There are many existing studies evaluating absorption models at frequencies below 200 GHz, for example Liljegren et al. (2005); Payne et al. (2008); Hewison (2006); Brogniez et al. (2016); Westwater et al. (2003); Cadeddu et al. (2007); Turner et al. (2009). However, validation of absorption models at the sub-millimetre frequencies used by ICI is currently limited. Experiments with ground-based systems in this frequency range are challenging due to the strong absorption from water vapour in the lower troposphere. As a result, it is only possible to make usable observations in extremely dry conditions such as at high-altitude sites. For example, studies by Mlawer et al. (2019) and Pardo et al. (2001) compared simulated brightness temperatures with spectrally-resolved observations from Fourier transform spectrometers at frequencies above 450 and 350 GHz respectively, and a recent study by Pardo et al. (2022) used extremely-high resolution measurements from the APEX astronomical observatory to evaluate an absorption model in the 578 to 738 GHz range.

Airborne radiometers can also be used to evaluate absorption models at sub-millimetre frequencies as they can make upward-looking observations from above the high concentrations of water vapour in the lower troposphere, as well as being able to approximate satellite viewing geometries with downward-looking observations from high altitudes. The International Sub-millimetre Radiometer (ISMAR, Fox et al., 2017) is an airborne demonstrator for ICI. It is flown on the UK's BAe-146-301 Atmospheric Research Aircraft (FAAM BAe-146) and in conjunction with the Microwave Airborne Radiometer Scanning System (MARSS, McGrath and Hewison, 2001) it covers the ICI frequency range, as well as including additional frequencies relevant to MWS and MWI. This study uses airborne observations from the MARSS and ISMAR radiometers to evaluate two selected atmospheric absorption models at frequencies between 89 and 664 GHz. It formed part of the EUMETSAT-funded project "Study on atmospheric absorption models using ISMAR data". A dataset of clear-sky airborne observations spanning a range of atmospheric conditions was collected, including radiometric observations from MARSS and ISMAR and associated measurements of atmospheric profiles of temperature and humidity. These data are used to perform radiative closure calculations with the two gas absorption models, both of which have been integrated into the Atmospheric Radiative Transfer Simulator (ARTS, Buehler et al., 2024) to determine their performance at the frequencies of interest. When performing radiative closure experiments it is important to consider the impact of uncertainties in the spectroscopic parameters on the simulated brightness temperatures. Cimini et al. (2018) showed how this could be applied to ground-based microwave radiometers, and a recent study by Gallucci et al. (2024), performed as part of the same EUMETSAT project, has extended the analysis to sub-millimetre frequencies and satellite and airborne viewing geometries.

The paper is organized as follows: the selected absorption models are described in sec. 2, and details of the airborne dataset are given in sec 3. Sec. 4 describes the radiative transfer simulations, the result of the closure calculations are discussed in sec. 5 and conclusions are presented in sec. 6.

**Table 1.** Comparison of updated AMSUTRAN and Ros22 absorption models.

| Feature | AMSUTRAN | Ros22 |
|---|---|---|
| $H_2O$ lines | 338 lines from AER v3.8 "fast" database | 20 most significant lines |
| Pressure shifts | Air-broadened | Air and self-broadened |
| $H_2O$ continuum | MT-CKD v3.5, frequency-dependent coefficients | Turner et al. (2009) coefficients, adjusted for new absorption line parameters |
| $O_2$ lines | Tretyakov et al. (2005), first-order line mixing | Makarov et al. (2020), second-order line mixing |
| $O_3$ lines | 652 lines from JPL v4 database | 464 lines from HITRAN-2020* |
| Line shape | Van-Vleck Weisskopf (VVW) | VVW with speed-dependent lineshape at 22.23, 118.75 and 183.31 GHz |
| Sources | Field campaigns to constrain key parameters | Majority from various laboratory studies |

* In this study we use the AMSUTRAN $O_3$ settings with both absorption models.

## 2 Absorption models

A review and comparison of absorption models applicable across the microwave and sub-millimetre spectral range was performed by Turner et al. (2022). This includes several commonly used absorption models, as well as the configuration of AMSUTRAN used for RTTOV v12 (described by Turner et al., 2019) and an updated version that was designed to improve its validity in the sub-millimetre spectral region. The updated version has been used to generate ICI coefficients for RTTOV v13, which also include measured ICI spectral response functions. Based on this study, we have selected two complete absorption models suitable for simulating ICI radiances. These are the updated AMSUTRAN configuration, which is effectively a line-by-line model that incorporates the AER water vapour spectroscopy, and the most recent (2022) iteration of the Rosenkranz (2017) model (referred to here as Ros22, and available from http://cetemps.aquila.infn.it/mwrnet/lblmrt_ns.html), which uses a significantly reduced set of spectral lines for computational efficiency. The AER and Rosenkranz models are actively maintained and developed and are expected to provide an acceptable framework for use within the 89 to 664 GHz frequency range. Both models incorporate the results of recent studies, and although they are based on similar principles they adopt different approaches to updating spectroscopic parameters. In particular, the AER model incorporates adjustments in response to atmospheric measurements from field experiments, and the Ros22 model is based strongly on laboratory studies. Both absorption models have been implemented within the ARTS radiative transfer model (Buehler et al., 2024) to allow them to be evaluated in a consistent manner against the airborne observations. The two absorption configurations are described in more detail below, and they are summarised in tab. 1.

The water vapour spectroscopy in the updated AMSUTRAN configuration now follows the AER model. Specifically, it uses version 3.8 of the "fast" line parameter database (Cady-Pereira et al., 2020), which contains 338 of the most significant water vapour lines below 1649 GHz (from a total of 1488 lines in the full AER list). The water vapour continuum uses version 3.5 of the semi-empirical MT-CKD model (Mlawer et al., 2012), which can be obtained from https://github.com/AER-RC/MT_CKD.

The line database is based on HITRAN-2012 (Rothman et al., 2013), but includes modifications to key parameters in response to measurements from atmospheric field campaigns (Mlawer et al., 2019). For consistency with the MT-CKD continuum the line absorption is calculated using a 750 GHz cut-off. The oxygen absorption in AMSUTRAN is unchanged and uses parameters from Tretyakov et al. (2005), which is also used by AER in the MonoRTM model. The dry-air continuum is also unchanged and is taken from Liebe et al. (1993). The original ozone absorption in AMSUTRAN only included 35 of the strongest ozone lines between 0 and 300 GHz with parameters taken from HITRAN-2000. The updated version includes the 652 ozone lines which are included in the AER "fast" line database. However, their parameters are taken from version 4 of the JPL line catalogue, with broadening parameters calculated following the standard HITRAN procedure from Wagner et al. (2002) with some adjustments (Iouli Gordon, Harvard & Smithsonian, personal communication, 2019). These line parameters are very similar to those included in the latest HITRAN-2020 release, but the line strengths are approximately 4% greater than the AER v3.8 and HITRAN-2016 values, with the latter being in error (Birk et al., 2019).

The Ros22 model has been developed to increase its suitability for frequencies up to 1000 GHz. The earlier 2017 version is described in detail by Cimini et al. (2018). The main differences between the 2017 and 2022 versions are the addition of water vapour lines at 658, 860, 970, 987 and 1097 GHz, the adjustment of broadening and shifting parameters for the 22 and 183 GHz lines (Tretyakov, 2016; Koshelev et al., 2018) and the addition of self-induced pressure shift parameters and pressure shift temperature dependencies. A speed-dependent line shape has been introduced at 22 and 183 GHz (Rosenkranz and Cimini, 2019; Koshelev et al., 2021), and a second-order approximation for oxygen line mixing is included (Makarov et al., 2020). Line parameters which are taken from the HITRAN database have been updated to values from the latest 2020 release (Gordon et al., 2022), and additional ozone lines up to 1 THz have also been included resulting in a total of 464 ozone lines. However, for simplicity this study uses the same ozone configuration for both the AMSUTRAN and Ros22 absorption models. Since there are only small differences between the JPL v4 and HITRAN-2020 ozone line parameters, the main differences will be caused by the slightly different subset of lines included in the two models, which is expected to have minimal impact. Note that we expect the uncertainty of the Ros22 model to be similar to the 2019 version evaluated by Gallucci et al. (2024).

## 3  Airborne dataset

A dataset of airborne observations suitable for performing radiative closure calculations was collated from clear-sky observations collected from the FAAM BAe-146 aircraft covering a wide range of atmospheric conditions. Radiometric observations from the MARSS (McGrath and Hewison, 2001) and ISMAR (Fox et al., 2017) radiometers cover the spectral region between 89 and 664 GHz. Table 2 lists the available channels, along with the closest-matching equivalent channels from MWI, MWS and ICI. MARSS and ISMAR are located on the side of the aircraft and are capable of along-track scanning in both upward and downward viewing directions. The MARSS antenna beamwidths are 11.8°, 11.0° and 6.2° full-width half-maximum at 89, 157 and 183 GHz respectively. All ISMAR beamwidths are less than 4°. Two viewing geometries are considered in this study. Downward-looking measurements during high-altitude flight segments provide the closest match to the satellite observations, but can be sensitive to surface properties in some channels. Due to the relatively warm surface radiative background they also

**Table 2.** Comparison of MARSS and ISMAR channels with MWI, MWS and ICI. "H", "V" and "mixed" refer to horizontal, vertical and mixed polarisation respectively.

| Instrument | Channel | Satellite instrument | Satellite channel | Feature |
|---|---|---|---|---|
| MARSS | 89 GHz (mixed) | MWI, MWS | 89 GHz (V & H) | Window |
| ISMAR | 118±1.1 GHz (V) | MWI | 118±1.2 GHz (V) | Oxygen |
| ISMAR | 118±1.5 GHz (V) | MWI | 118±1.4 GHz (V) | Oxygen |
| ISMAR | 118±2.1 GHz (V) | MWI | 118±2.1 GHz (V) | Oxygen |
| ISMAR | 118±3.0 GHz (V) | MWI | 118±3.2 GHz (V) | Oxygen |
| ISMAR | 118±5.0 GHz (V) | | | Oxygen |
| MARSS | 157 GHz (H) | MWI, MWS | 165.5 GHz (V) | Window |
| MARSS | 183±1.0 GHz (H) | ICI | 183±2.0 GHz (V) | Water vapour |
| | | MWS | 183±1.0 GHz | |
| MARSS | 183±3.0 GHz (H) | ICI | 183±3.4 GHz (V) | Water vapour |
| | | MWS | 183±3.0 GHz | |
| MARSS | 183±7.0 GHz (H) | ICI | 183±7.0 GHz (V) | Water vapour |
| | | MWS | 183±7.0 GHz | |
| ISMAR | 243 GHz (V & H) | ICI | 243 GHz (V & H) | Window |
| | | MWS | 229 GHz | |
| ISMAR | 325±1.5 GHz (V) | ICI | 325±1.5 GHz (V) | Water vapour |
| ISMAR | 325±3.5 GHz (V) | ICI | 325±3.5 GHz (V) | Water vapour |
| ISMAR | 325±9.5 GHz (V) | ICI | 325±9.5 GHz (V) | Water vapour |
| ISMAR | 448±1.4 GHz (V) | ICI | 448±1.4 GHz (V) | Water vapour |
| ISMAR | 448±3.0 GHz (V) | ICI | 448±3.0 GHz (V) | Water vapour |
| ISMAR | 448±7.2 GHz (V) | ICI | 448±7.2 GHz (V) | Water vapour |
| ISMAR | 664 GHz (V & H) | ICI | 664 GHz (V & H) | Window |

have a lower sensitivity to the atmospheric absorption compared to upward-looking views with a cold space background. As discussed by Hewison (2006), vertical profiles of upward-looking brightness temperatures (i.e. measurements of zenith brightness temperatures made from many different altitudes) have a strong sensitivity to the absorption model, and compared to ground-based observations they can provide data covering a wide range of temperatures and pressures.

The dataset contains observations from ten flights performed around the UK between March 2015 and August 2021 that specifically targeted clear sky observations with MARSS and ISMAR. These flights all contain at least one "stepped spiral descent" that was used to obtain a vertical profile of brightness temperatures. An example is shown in fig. 2, where the aircraft track consists of a series of straight and level runs lasting approximately two minutes each, starting at high altitude and separated in height by 2000 ft ( 600 m). The aircraft turned during the descending section between each run to remain within a relatively

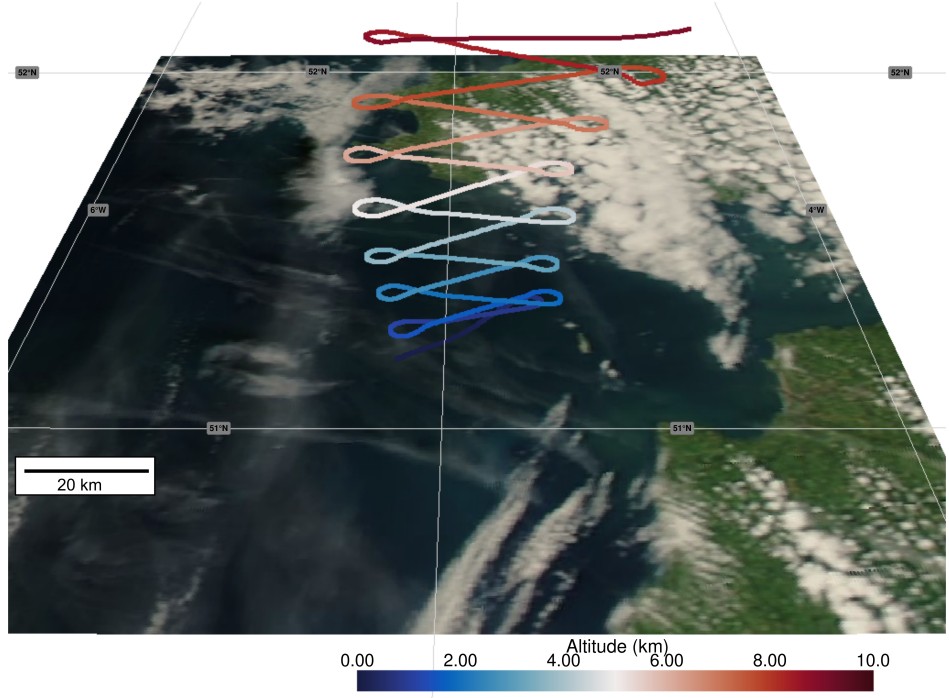

**Figure 2.** Example aircraft track from a stepped spiral descent during flight C246. The aircraft track is shown between 12:37 and 13:44 UTC and the background MODIS image is taken from the Aqua overpass at 13:51 UTC (imagery provided by services from NASA's Global Imagery Browse Services (GIBS)).

compact operating area with a horizontal scale of approximately 50 km. Measurements during the level runs can be averaged to reduce the impact of noise. A complete spiral descent from maximum altitude to near-surface takes approximately 60 minutes to complete. Eight flights also contained a high-altitude leg (between 8.5 and 10.3 km) flown above the sea during which downward-looking measurements were made, and dropsondes were released during seven of these flights to measure the vertical profile of temperature and humidity below the aircraft. Where available, downward-looking observations were cloud-screened using co-located profiles from the Leosphere ALS-450 Lidar system on board the FAAM aircraft. A small number of additional observations were also removed, where visual observations noted the presence of low cloud in the area and the brightness temperatures at 89 GHz were enhanced by ~2 K compared to the rest of the run.

An additional three opportunistic flights were also included in the dataset to increase the range of atmospheric conditions that were sampled, including one Arctic flight and one tropical flight . These flights contain clear-sky observations from either a high-altitude run or a continuous profile descent, but the tropical flight does not include any ISMAR observations. The continuous profiles have a larger horizontal extent than the stepped spirals, and since they do not contain level runs it is not possible to average the radiometric observations to reduce the impact of noise. However, Hewison (2006) demonstrated that they can still provide useful observations for absorption model validation. Intermittent problems with several of the ISMAR

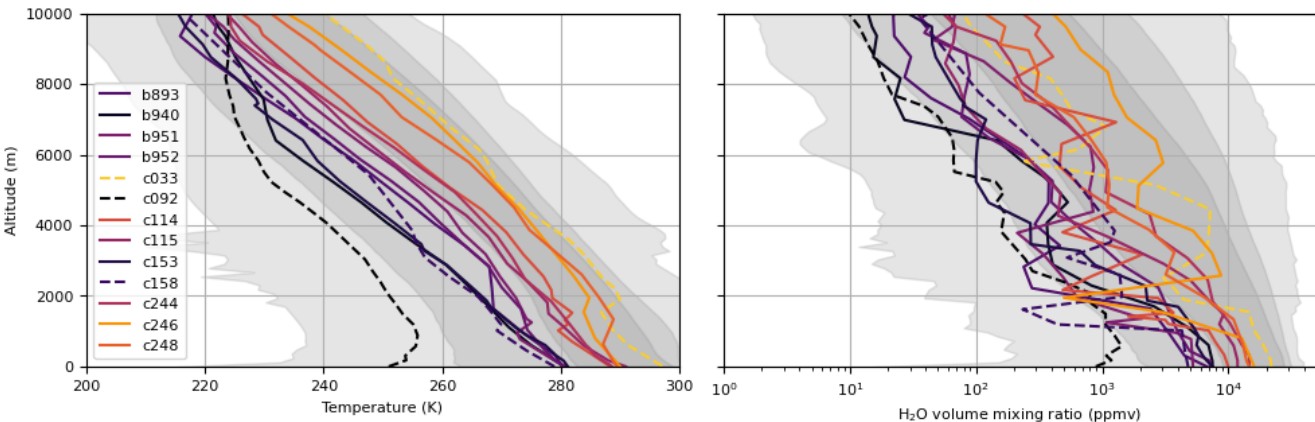

**Figure 3.** Atmospheric temperature and humidity profiles from FAAM flights. The solid lines represent the ten targeted flights, and the dashed lines represent the three opportunistic flights. The gray shading shows the range of profiles from the dataset of Eresmaa and McNally (2014) designed to give diverse sampling of specific humidity; from lightest to darkest these represent the minimum/maximum, 5th-95th percentile and 25th-75th percentile respectively.

receivers meant that not every channel was available on each flight. A summary of the different flights contributing to the dataset, including which channels were operational, can be found in tab. 3.

The vertical profiles of atmospheric temperature and humidity required as inputs to the radiative closure calculations were measured using aircraft in-situ instruments during profile descents. Temperatures were measured using platinum resistance sensors housed in a Rosemount Aerospace Inc. Type 102 Total Temperature Housing. Humidity measurements were made using the WVSS-II, Buck CR2 and General Eastern 1011B hygrometers which are described by Vance et al. (2015), although not all of the instruments were flown on every flight. A manual assessment of the in-situ profiles was performed, considering

the characteristics of the different instruments, to derive a best estimate for the temperature and humidity at each altitude in the stepped spiral profile, as well as plausible ranges of maximum and minimum temperatures and humidities based on both the variability sampled along the runs and the differences between the measurements. The individual profiles are plotted in fig. S1 in the supplement. Dropsondes were released during all but one of the high-altitude runs to provide measurements of the atmospheric profile below the aircraft. The majority of the measurements used Vaisala RD94 sondes. During flights C244 and

C246 the more recent RD41 sonde type was used, and both sonde types were deployed during flight C248. All sondes were processed using the ASPEN software. Direct comparisons have since been performed between the two sonde types and show a dry bias in the older RD94 sondes, probably caused by contamination of the humidity sensor due to long storage times. To compensate for this the humidity mixing ratio measured by the RD94 sondes was increased by 15%.

Figure 3 shows the atmospheric profiles sampled by the aircraft during each flight compared to the range of values from the

dataset of Eresmaa and McNally (2014) which has been designed to give a representative sample of global diversity. Here, the data selected to maximise the diversity of specific humidity has been used. Due to the larger number of flights in UK winter

**Table 3.** Summary of flights contained in the dataset used for this study. (cp - continuous profile, ✓ - substantially complete data from channel, ○ - partial data from channel)

| Flight | Date | Location | No. of stepped spirals | High-altitude run | IWV (kgm$^{-2}$) | 89 | 118 ±1.1 | 118 ±1.5 | 118 ±2.1 | 118 ±3.0 | 118 ±5.0 | 157 | 183 ±1.0 | 183 ±3.0 | 183 ±7.0 | 243 H | 243 V | 325 ±1.5 | 325 ±3.5 | 325 ±9.5 | 448 ±1.4 | 448 ±3.0 | 448 ±7.2 | 664 H | 664 V |
|---|---|---|---|---|---|---|---|---|---|---|---|---|---|---|---|---|---|---|---|---|---|---|---|---|---|
| B893 | 10 Mar 2015 | UK | 1 | Yes | 5.9 | ✓ | ✓ | ✓ | ✓ | ✓ | ✓ | ✓ | ✓ | ✓ | ✓ | ✓ | ✓ | ✓ | ✓ | ✓ | ✓ | ✓ | ✓ | ✓ | ✓ |
| B940 | 10 Feb 2016 | UK | 1 | Yes | 8.5 | ✓ | ✓ | ✓ | ✓ | ✓ | ✓ | ✓ | ✓ | ✓ | ✓ | ✓ | ✓ | ✓ | ✓ | ✓ | ✓ | ✓ | ✓ | ✓ | ✓ |
| B951 | 16 Mar 2016 | UK | 1 | No | 6.5 | ✓ | ✓ | ✓ | ✓ | ✓ | ✓ | ✓ | ✓ | ✓ | ✓ | ○ | ✓ | ○ | ○ | ✓ | ✓ | ✓ | ✓ | ○ | ✓ |
| B952 | 17 Mar 2016 | UK | 1 | Yes | 8.8 | ✓ | ✓ | ✓ | ✓ | ✓ | ✓ | ✓ | ✓ | ✓ | ✓ | ○ | ✓ | ○ | ✓ | ○ | ○ | ○ | ○ | ✓ | ✓ |
| C033 | 22 Aug 2017 | Tropical Atlantic | cp | No | 31.4 | ✓ |  |  |  |  |  |  | ✓ | ✓ | ✓ |  | ✓ |  |  |  |  |  |  |  |  |
| C092 | 22 Mar 2018 | NWT Canada | cp | No | 2.3 | ✓ | ✓ | ✓ | ✓ | ✓ | ✓ | ✓ | ✓ | ✓ | ✓ | ✓ | ✓ | ○ | ✓ | ✓ | ✓ | ✓ | ✓ | ✓ | ✓ |
| C114 | 05 Sep 2018 | UK | 1 | Yes | 11.3 | ✓ | ✓ | ✓ | ✓ | ✓ | ✓ | ✓ | ✓ | ✓ | ✓ | ✓ | ✓ |  | ✓ | ✓ | ✓ | ✓ | ✓ | ○ | ○ |
| C115 | 06 Sep 2018 | UK | 1 | No | 17.2 | ✓ | ✓ | ✓ | ✓ | ✓ | ✓ | ✓ | ✓ | ✓ | ✓ | ○ | ✓ | ○ | ○ | ✓ | ✓ | ✓ | ✓ | ○ | ○ |
| C153 | 13 Mar 2019 | UK | 1 | Yes | 8.8 | ✓ | ✓ | ✓ | ✓ | ✓ | ✓ | ✓ | ✓ | ✓ | ✓ | ○ | ✓ | ✓ | ✓ | ✓ | ○ | ○ | ○ | ✓ | ✓ |
| C158 | 18 Mar 2019 | UK |  | Yes | 6.4 | ✓ | ✓ | ✓ | ✓ | ✓ | ✓ | ✓ | ✓ | ✓ | ✓ | ✓ | ✓ | ✓ | ✓ | ✓ | ✓ | ✓ | ✓ | ✓ | ✓ |
| C244 | 07 Jul 2021 | UK | 2 | Yes | 23.1 | ✓ | ✓ | ✓ | ✓ | ✓ | ✓ | ✓ | ✓ | ✓ | ✓ | ✓ | ✓ | ○ | ○ | ○ | ○ | ✓ | ○ | ○ | ○ |
| C246 | 14 Jul 2021 | UK | 2 | Yes | 24.1 | ✓ | ✓ | ✓ | ✓ | ✓ | ✓ | ✓ | ✓ | ✓ | ✓ | ✓ | ○ | ○ | ○ | ✓ | ✓ | ○ | ○ | ○ | ○ |
| C248 | 26 Aug 2021 | UK | 2 | Yes | 21.8 | ✓ | ✓ | ✓ | ✓ | ✓ | ✓ | ✓ | ✓ | ✓ | ✓ | ✓ | ✓ | ○ | ○ | ○ | ○ | ○ | ○ | ○ | ✓ |

conditions, the aircraft sampling is biased towards colder, drier profiles, but there is reasonable coverage between the 5th and 75th percentiles of both temperature and humidity. Note that although there are no ISMAR measurements from the single opportunistic tropical flight (C033), the highest water vapour concentrations in the upper troposphere were encountered during UK summer flights. The most extreme warm/moist profiles, which are not covered by the airborne dataset, are likely to be associated with tropical convection and will therefore often occur in cloudy conditions. The column-integrated water vapour (IWV) for each profile, calculated from the aircraft in-situ measurements, is reported in tab. 3 and covers the range 2.3 - 31.4 $\mathrm{kgm}^{-2}$.

## 4  Radiative transfer simulations

The radiative closure was performed by comparing the observed brightness temperatures to simulated values using the atmospheric profiles. The simulations were performed using version 2.5.11 of the ARTS radiative transfer model (Buehler et al., 2024). Clear sky simulations were performed for a 1-dimensional atmosphere. To minimise errors associated with the spatial discretisation of the radiative transfer equation the maximum propagation path length was set to 250 m. For downward-looking calculations the sea surface emissivity was modelled using TESSEM2 (Prigent et al., 2016). The surface temperature was taken from infrared measurements made during low-level aircraft runs, and the near-surface wind speed was measured by the dropsondes. The profiles of temperature, pressure and humidity were extended above the maximum altitude sampled by the aircraft or dropsondes using values from the Met Office global NWP model, and the ozone profile was taken from the ERA-5 reanalysis.

The simulations covered the frequency ranges of the MARSS and ISMAR channels with a resolution of 25 MHz. Tests at a finer frequency resolution of 3 MHz showed differences less than 0.06 K. For ISMAR, the frequency-resolved simulations were convolved with the measured channel spectral response functions (SRFs). For zenith simulations this can lead to differences of up to 3 K compared with assuming an ideal SRF that is uniform across the channel passband, although for most channels the difference is less than 1 K. The largest differences occur for channels centred around the 118 and 325 GHz absorption lines where the SRF has a significant spectral slope, as this shifts the location of the effective centre of the passband. Note that measured SRFs are not available for MARSS and an idealised uniform SRF is assumed.

For the MARSS window channels, which have rather wide main beams, it is also necessary to include the finite antenna beamwidth in the simulations. For nadir simulations in these channels the finite width of the main beam leads to differences up to approximately 0.2 K compared to an ideal "pencil beam". The beamwidth is modelled using the ARTS capability to include a 1-dimensional Gaussian antenna pattern. Internally, this calculates "pencil beam" radiances at multiple angles which are then weighted accordingly. No correction is applied for ISMAR nadir simulations as tests showed that, due to the narrower beamwidth, the difference is less than 0.1 K. For zenith simulations the finite beamwidth leads to differences of up to 0.7 K for the largest values of IWV at 157 GHz. However, this is not modelled in ARTS in order to reduce the computational time associated with simulating radiances at multiple angles. Instead, the correction described by Han and Westwater (2000) (eq. 30) is applied to correct the simulated brightness temperatures in all channels. Wide antenna beams could also introduce errors due

to atmospheric and surface inhomogeneities across the antenna footprint. However, because of the low altitude of the aircraft compared to a satellite, even the widest beams have a ground footprint of approximately 2 km and we do not expect significant spatial inhomogeneities on these length-scales.

## 5    Results and discussion

### 5.1    Upward-looking radiative closure

Upward-looking brightness temperatures were simulated for all run altitudes during the spiral descents for each flight using the best-estimate atmospheric profiles that were derived from the aircraft in-situ measurements as described in sec. 3. The difference between the observed and simulated brightness temperatures using the AMSUTRAN absorption model are shown in fig. 4 as a function of the partial column water vapour, i.e. the column-integrated mass of water vapour above the aircraft. This parameter was selected instead of the run altitude or pressure to give a degree of normalisation between flights with very

different water vapour amounts, although it is perhaps less appropriate for the channels around 118 GHz which are dominated by oxygen absorption. Also shown in the figure is the range of brightness temperatures simulated using the extreme warm/wet and cold/dry atmospheric profile estimates from the in-situ measurement, which indicates the range of brightness temperatures that might be expected due to atmospheric variability.

   The figure shows that there can be considerable differences, in some cases greater than 20 K, between observations and

simulations using the best-estimate atmospheric profiles. The differences are smallest for the low-frequency window channel at 89 GHz and the channels centred on the 118 GHz oxygen line. Since these are the channels which are least sensitive to water vapour, this suggests that the larger differences seen in the other channels are strongly influenced by uncertainties in the water vapour profile. The large variability between the flights in both the sign and magnitude of the differences means they are unlikely to be caused by systematic errors in the measurements or simulations, and in most cases the observed differences lie

within the range of simulations made using the observed extreme warm/wet and cold/dry profiles.

   Focusing on the water vapour channels, for a given flight there are clear correlations between the brightness temperature differences for channels at different frequencies but with similar sensitivity to water vapour. This suggests that a large contribution to the differences comes from the representativity of the atmospheric water vapour profile used in the simulations. Water vapour can be highly variable, even over the relatively compact area sampled by the spiral descents, meaning that the

best-estimate in-situ measurements may not adequately represent the profiles at the time and location of the radiometric observations. This is further demonstrated by the large spread of simulated brightness temperatures when using the atmospheric profiles based on the extremes of the in-situ observations, which is generally larger than the brightness temperature differences between observations and simulations. This highlights the importance of accurately measuring the water vapour profile across the field of view of the instrument at the time of measurement when performing such closure studies, although this is difficult

to achieve in practice.

   The impact of the representativity error can be reduced by calculating the mean differences across multiple flights. The brightness temperature differences were grouped into ten bins with equal numbers of points based on the partial column

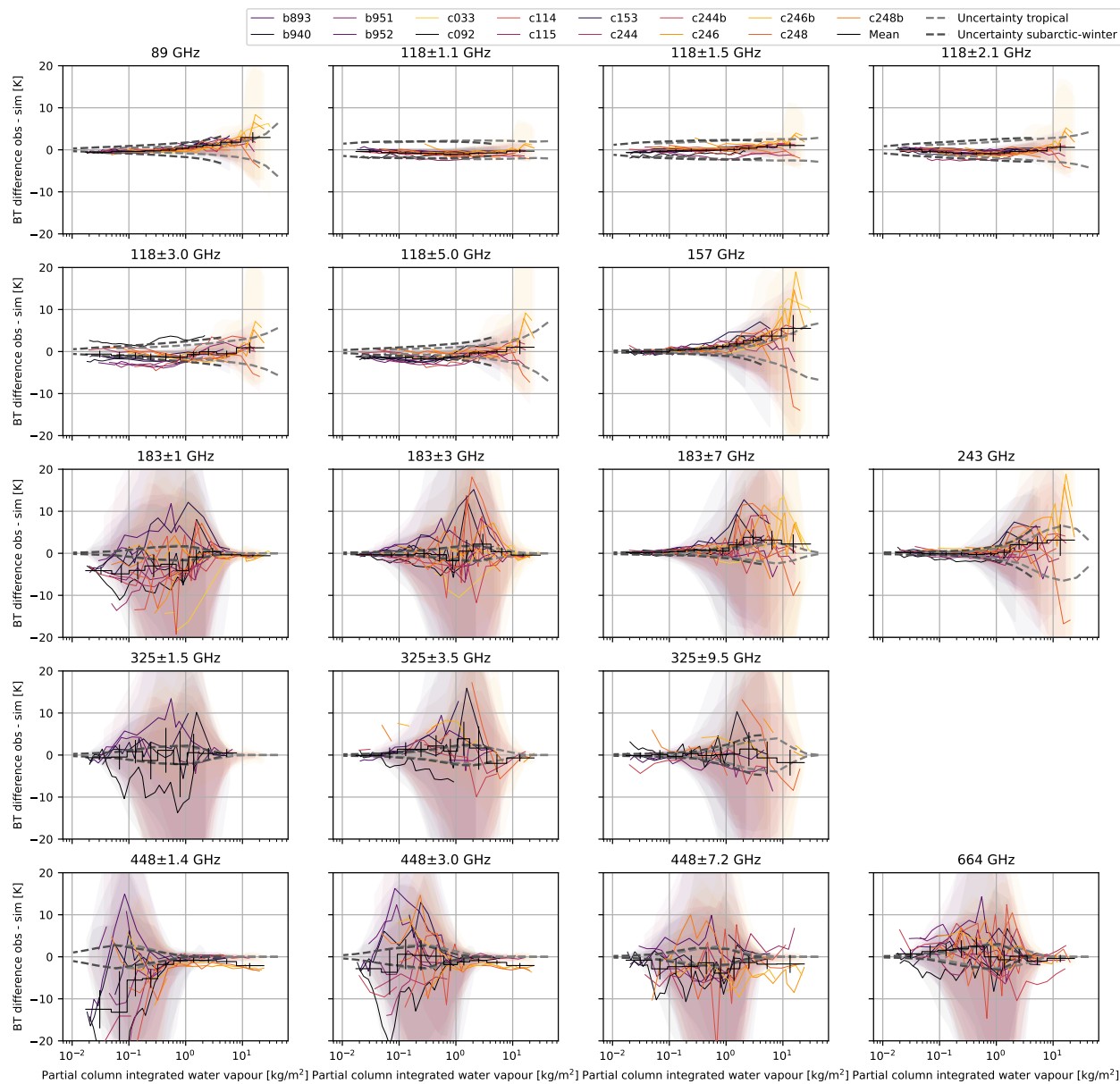

**Figure 4.** Difference between observed and simulated brightness temperatures using the AMSUTRAN absorption model for upward-looking views during vertical profiles, as a function of the partial column water vapour (column-integrated water vapour mass above the aircraft). The coloured lines represent the simulations using the best-estimate in-situ atmospheric profiles, and the background shading indicates the range of simulated brightness temperatures using the extreme warm/wet and cold/dry profiles for each flight. The stepped black line shows the mean difference across all flights with the error bars indicating the 95% confidence interval (CI) for the mean. The gray dashed lines show the 95% CI (2-$\sigma$) spectroscopic uncertainty from Gallucci et al. (2024) for representative tropical and sub-arctic winter profiles. Larger versions of the individual panels are available in fig. S2 in the supplementary material.

water vapour, and the bin-mean values and 95% confidence interval (CI) for the mean, based on the spread and number of values within the bin, were calculated. These are also shown in Fig 4, as well as the spectroscopic uncertainty from Gallucci et al. (2024) for tropical and sub-arctic winter profiles. This is the uncertainty in simulated brightness temperatures caused by imperfect knowledge of the spectroscopic parameters in the absorption model, and here we show a 2-$\sigma$ uncertainty, equivalent to the 95% CI. The largest uncertainties are seen in the window channels, where the dominant source is the water vapour continuum. At 89 GHz the oxygen line mixing parameters also have a significant impact. The spectroscopic uncertainty for the two profiles for a given partial column water vapour is generally similar, although the tropical profile extends to larger values. The small differences in spectroscopic uncertainty are mainly due to the uncertainty in the temperature dependence of the parameters, which has a greater impact for the colder sub-arctic profile.

Although the sample size is relatively small, the flight mean brightness temperature differences are within, or close to, the spectroscopic uncertainty for most cases. The biggest systematic deviations occur at 183$\pm$1 and 448$\pm$1.4 GHz for very low values of partial column water vapour. These channels are close to the centre of water vapour absorption lines and are sensitive to low water vapour concentrations which are challenging to measure with the available in-situ instruments. They are also sensitive to the small amounts of water vapour above the maximum altitude sampled by the aircraft, where the atmospheric profiles are taken from the NWP model and do not have associated wet and dry extremes.

Note that the figure does not show the uncertainties due to errors in the radiometric observations. Since typically around thirty measurements are averaged at each run altitude to create the mean brightness temperature profiles, the impact of random radiometric noise (i.e. NE$\Delta$T) is less than 0.5 K even for the noisiest ISMAR channels which have an NE$\Delta$T of up to approximately 3 K when measuring very cold brightness temperatures (Fox et al., 2017). Systematic biases in the ISMAR measurements are also discussed by Fox et al. (2017), and are mainly caused by uncertainties in the effective radiometric temperatures of the calibration black-bodies, with some channels also impacted by standing wave effects (118$\pm$3 and 664 GHz) which can cause an additional slowly-varying systematic error. The bias is dependent on the scene temperature, and is largest when viewing scenes that are significantly colder than either of the on-board calibration targets. Neglecting the standing waves, the maximum estimated bias is around 3 K. Although the standing wave effects that are present in some channels increase the worst-case bias estimate, they are not expected to be consistent between flights so the average impact will be reduced. This is also a worst-case estimate, and in practice the accuracy is likely to be significantly better than 3 K. Any bias due to uncertainties in the calibration target temperatures will be correlated between channels which share a common receiver front-end, and such patterns are not obviously apparent in the flight-mean results shown in fig 4.

An additional uncertainty also arises from the choice of radiative transfer model. Even with consistent spectroscopy, different models do not produce identical results, mainly due to the method used to discretise the radiative transfer equation and the assumptions within the model on how the atmospheric profiles vary between the discrete vertical levels provided as inputs. The differences between models generally reduces as the vertical resolution is increased. Melsheimer et al. (2005) compared the version of ARTS available at the time to three other models for downward-looking AMSU-B simulations and found that, when consistent spectroscopy was implemented across the models, differences were less than ~1K with the exception of one outlying

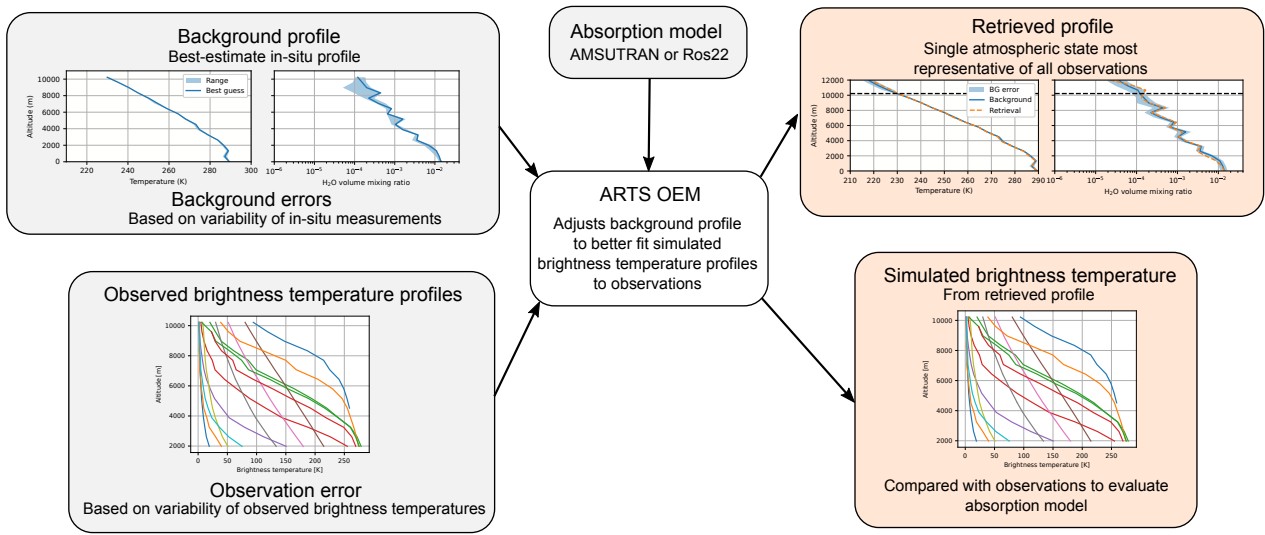

**Figure 5.** Overview of the OEM retrieval method used to reduce the impact of profile representativity errors.

model at 183 GHz. Much smaller differences (~0.1K) were seen for upward-looking simulations. We have tried to minimise the uncertainty due to the radiative transfer model in this study by performing calculations with a high vertical resolution.

Another approach to reducing the impact of errors in the representativity of the in-situ observations of the atmospheric profile
is to use the radiometric observations to retrieve the atmospheric state. For a given gas absorption model this method searches for an atmospheric profile that best matches the measured brightness temperature profiles. An overview of the retrieval method is given in fig. 5. The in-situ profile was taken as the a-priori background state, and the optimal estimation method (OEM) was used to simultaneously adjust the temperature and water vapour profiles to give a better match between the observed and simulated brightness temperatures using each absorption model. The ability of the retrieval to fit the observations across all
frequencies is an indication of the accuracy of the gas absorption model. In the retrievals presented here, the mean brightness temperatures observed at each of the altitudes sampled by level runs during the stepped spiral descent are used simultaneously to retrieve a single atmospheric state. This means the brightness temperatures provide a strong constraint on the retrieved atmospheric profile, but it assumes that the atmosphere is sufficiently homogeneous both spatially and temporally that it can be represented by a single profile. Note that the $118\pm3$ GHz channel was excluded from the retrieval as it has potentially
significant calibration biases due to standing wave effects which could result in undesirable large perturbations to the retrieved temperature profile.

Within the retrieval, observation error is taken as the standard deviation of the observed brightness temperatures along each run, assuming no correlations between channels. A minimum threshold of 1 K is applied for runs with very low observed variance. The background error is based on the estimated extreme warm (wet) and cold (dry) temperature and humidity profiles
from the in-situ observations. The errors are set to half the difference between the warm (wet) and cold (dry) extremes. The temperature errors are limited to the range 0.25-2.5 K, and the humidity errors, which are expressed in relative units, are

limited to the range 0.2-0.5. For heights above the maximum altitude sampled by the aircraft, where there are no estimates of extreme values, the maximum limits for temperature and humidity are used. To reduce oscillations in the retrieved profiles, the background error for water vapour was assumed to have a Gaussian correlation between vertical levels, using a 1 km scale height.

Figure 6 compares the observed brightness temperatures with the simulated brightness temperatures from the retrieved atmospheric profiles using the AMSUTRAN absorption model. This demonstrates how well the retrieval was able to fit the observed brightness temperatures. Compared to fig. 4 the brightness temperature differences are significantly smaller, and the variability between flights is also significantly reduced. This suggests that the retrieval is capable of reducing the representativity errors of the best-estimate in-situ profiles. In many cases the brightness temperature differences lie close to, or within, the theoretical estimate of the spectroscopic uncertainty. A notable outlier in fig. 6 is flight C114, which shows large oscillations, particularly at 183±1, 183±3, 448±7.2 and 664 GHz, with similar patterns also visible in some other channels. This flight encountered horizontal gradients in water vapour, such that the offset in aircraft track between adjacent levels in the profile resulted in alternately wetter and drier airmasses being sampled. The oscillations in the brightness temperature differences result from the retrieval attempting to fit a single water vapour profile to all the brightness temperature observations simultaneously.

Although the retrieval gives a better fit to the observed brightness temperatures it is difficult to show that the retrieved profile is accurate. However, fig. S4 in the supplement compares the retrieved and a-priori background profiles, and shows that in most cases the adjustments made by the retrieval are small and within the observed atmospheric variability. The largest changes are predominantly above the maximum altitude sampled by the aircraft, where the background state is taken from an NWP model. The fact that the retrieval is able to simultaneously improve the fit to observations across multiple absorption lines is also an indication that errors in the atmospheric profile are a main cause of the difference between observation and simulation, because any errors in the absorption model parameters should not be strongly correlated between the different lines.

Comparing the flight-mean results between fig. 4 and fig. 6 there are some differences. The bias observed at channels close to the centre of the water vapour lines at 183±3 and 448±1.4 GHz for low partial column water vapour is significantly reduced by the retrievals. This is because the retrievals produce systematically lower amounts of water vapour, particularly at heights above the maximum altitude sampled by the aircraft where the best-estimate profile is taken from NWP model data. Note that a similar effect is not seen at 325±1.5 GHz, which is also close to the centre of a water vapour line, simply because this channel was not available during the flights which contribute most strongly to the bias. This suggests that there may be a wet bias in the upper tropospheric/lower stratospheric water vapour within the NWP model, although it is not possible to completely exclude deficiencies in the absorption model or measurement biases. However, the fact that a similar effect is seen on two different absorption lines, and on measurements from independent instruments, suggests that these are less likely to be the cause. There is also a reduction in the bias in the window channels, particularly at 157 and 243 GHz, and also at 183±7 GHz, for large values of partial column water vapour. This is caused by the retrieval slightly increasing the humidity in the vicinity of the top of the boundary layer. Since the in-situ profile is used with a vertical resolution of 2000 ft ( 600 m), it is not able to capture the details of the strong gradients in water vapour expected around the top of the boundary layer, and the precise vertical location of these changes can have a large impact on brightness temperatures measured near the boundary layer top.

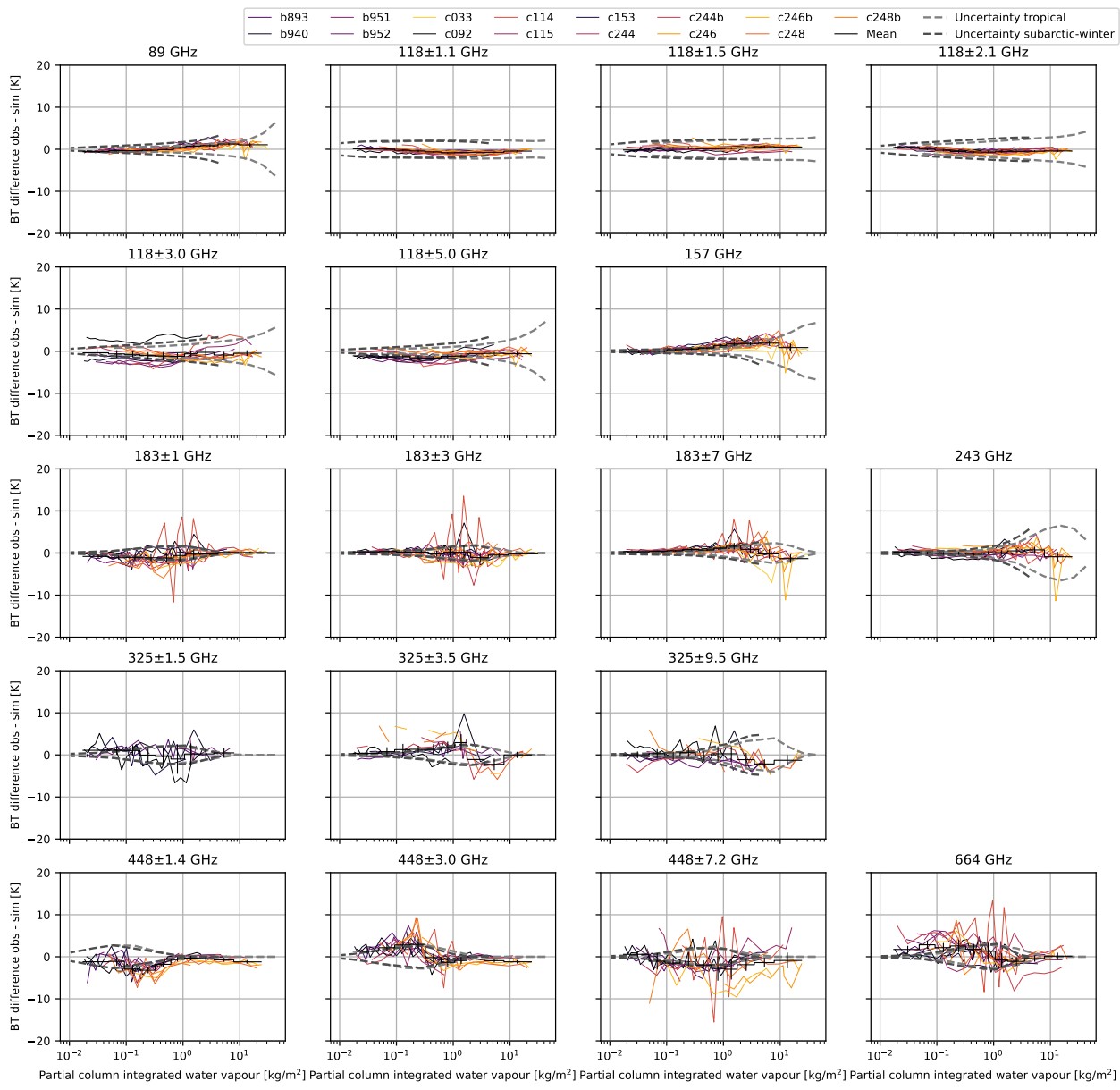

**Figure 6.** As fig. 4 using the retrieved atmospheric profiles and AMSUTRAN absorption model. Larger versions of the individual panels are available in fig. S3 in the supplementary material.

The upward-looking radiative closure results using both the AMSUTRAN and Ros22 absorption models are shown in fig. 7, which compares the flight-mean results for both best-estimate and retrieved atmospheric profiles using the two models. The figure shows that the AMSUTRAN and Ros22 models give very similar results, although there are a few differences in the details, for example the Ros22 model provides a slightly better fit to the low-frequency window channels at 89 and 157 GHz for both the best-estimate and retrieved profiles, and the AMSUTRAN model perhaps provides a better fit at 183±3 and 664 GHz for intermediate values of partial column water vapour. The mean absolute deviation (MAD) and bias of the binned flight-mean brightness temperature differences are listed in tab. 4. Since the bias can be small in cases where there are compensating positive and negative errors at different values of IWV, the MAD is our preferred statistic for overall comparison between the models. For the best-guess profiles the MAD is less than 2 K for both absorption models, with the exception of 183±1 and 448±1.4 GHz which, as discussed above, are strongly affected by the small amounts of water vapour above the maximum altitude sampled by the aircraft, and 448±7.2 GHz. The retrieval reduces the MAD compared to the best-guess profiles for almost all channels, and results in a maximum MAD of 1.6 K for the AMSUTRAN model at 448±7.2 GHz, with many channels having significantly smaller MAD. The mean value of the MAD across all channels is 1.31 K (1.41 K) for the AMSUTRAN and (Ros22) models respectively using the best-guess profiles, reducing to 0.84 K (0.85 K) for the retrieved profiles. Considering only the channels relevant for ICI, i.e. with frequencies of 183 GHz or greater, slightly increases the channel-mean MAD to 1.58 K (1.70 K) for the best-guess profiles, and 0.95 K (0.95 K) for the retrieved profiles. The AMSUTRAN and Ros22 models therefore have very similar performance, and both are suitable for use in the ICI frequency range.

## 5.2 Downward-looking radiative closure

The upward-looking radiative closure results discussed in the previous section provide a relatively sensitive test of the atmospheric absorption models due to the well-defined cold radiative background and the wide range of integrated water vapour that can be sampled by measuring at different altitudes. However, it is also of interest to consider downward-looking observations measured from high altitude as this gives the closest match to the satellite viewing geometry. Radiative closure calculations using downward-looking observations will be used to validate the radiometric accuracy of ICI during the post-launch cal/val period. During nine flights the aircraft performed a high altitude run as shown in tab. 3. These runs were performed at altitudes between 8.5 and 10.3 km. During eight of the flights dropsondes were released to measure the atmospheric profile below the aircraft in addition to the aircraft in-situ profiles measured before and/or after the run. A total of 33 dropsondes were released, although they were not uniformly distributed between the flights.

Figure 8 shows an example of the observations and nadir simulations using the AMSUTRAN absorption model from the high altitude run during flight B893. The simulations were performed using atmospheric profiles from the dropsondes, aircraft in-situ measurements and NWP model fields from the Met Office 1.5 km resolution operational forecast model at a range of lead times. For the dropsonde simulations the surface temperature was taken from infrared measurements made when the aircraft was flying at low altitude, and the near-surface wind speed, which is needed to model the surface emissivity, was measured by the dropsonde. Simulation using the aircraft in-situ profiles were performed for the best-estimate and extreme wet and dry profiles. For these simulations the wet and dry profiles are used with the coldest and warmest temperature profiles

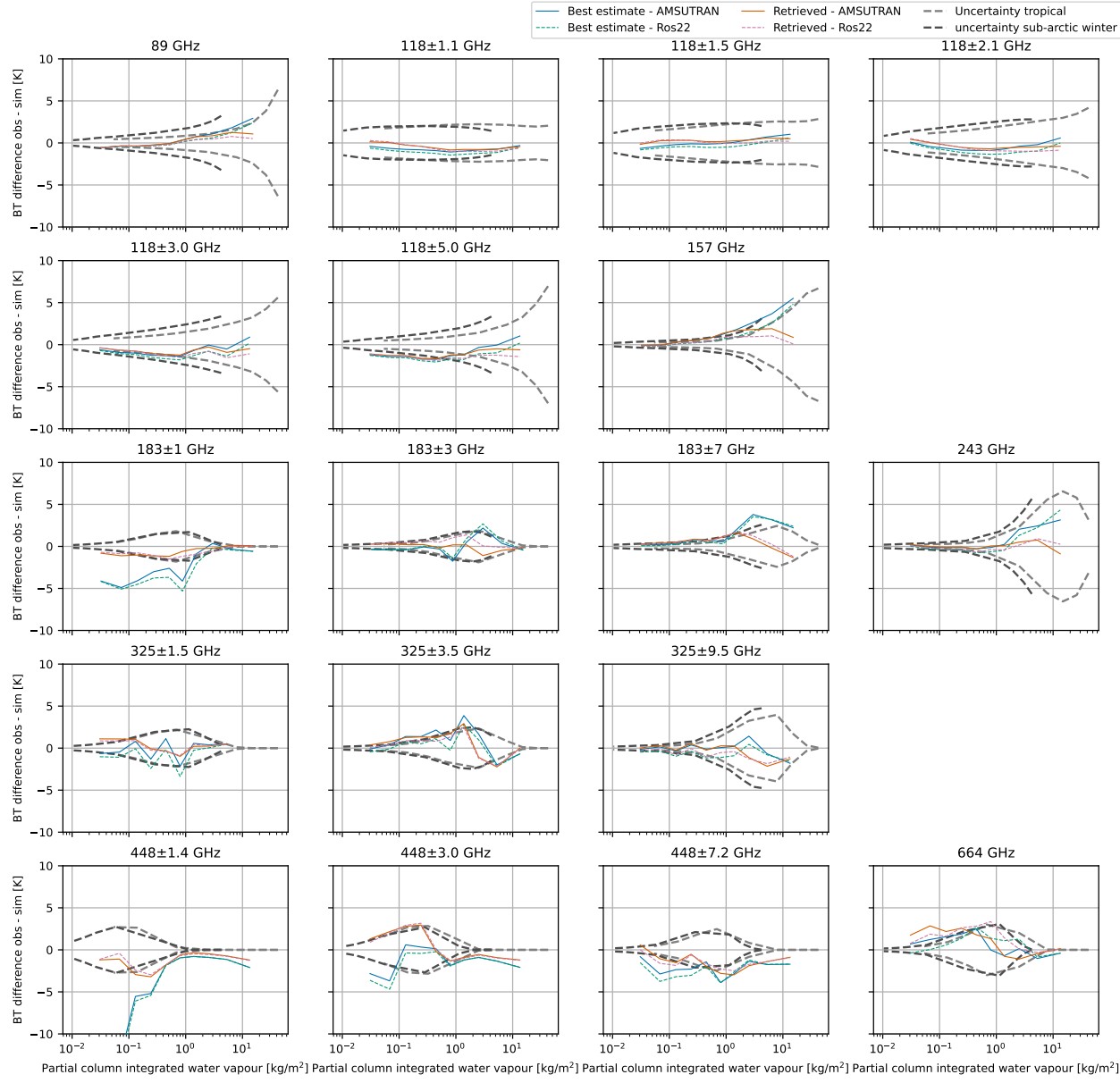

**Figure 7.** Mean differences between observed and simulated upward-looking brightness temperatures across all flights for the AMSUTRAN and Ros22 absorption models using best-estimate in-situ and retrieved atmospheric profiles.

**Table 4.** Mean absolute deviation (MAD) and bias of binned flight-mean upward-looking brightness temperature differences for best-guess and retrieved profiles using the AMSUTRAN and Ros22 absorption models.

| | MAD | | | | Bias | | | |
| --- | --- | --- | --- | --- | --- | --- | --- | --- |
| | Best guess | | Retrieved | | Best guess | | Retrieved | |
| | AMSUTRAN | Ros22 | AMSUTRAN | Ros22 | AMSUTRAN | Ros22 | AMSUTRAN | Ros22 |
| 89 GHz | 0.85 | 0.67 | 0.58 | 0.42 | 0.53 | 0.28 | 0.27 | 0.05 |
| 118±1.1 GHz | 0.75 | 1.07 | 0.50 | 0.63 | -0.75 | -1.07 | -0.47 | -0.55 |
| 118±1.5 GHz | 0.37 | 0.47 | 0.31 | 0.15 | 0.06 | -0.33 | 0.27 | 0.14 |
| 118±2.1 GHz | 0.57 | 0.86 | 0.46 | 0.69 | -0.44 | -0.86 | -0.36 | -0.57 |
| 118±3.0 GHz | 0.86 | 1.15 | 0.75 | 1.02 | -0.69 | -1.12 | -0.75 | -1.02 |
| 118±5.0 GHz | 1.14 | 1.40 | 1.11 | 1.45 | -0.94 | -1.37 | -1.11 | -1.45 |
| 157 GHz | 1.67 | 1.18 | 0.94 | 0.50 | 1.64 | 1.10 | 0.92 | 0.44 |
| 183±1 GHz | 2.50 | 2.98 | 0.66 | 0.76 | -2.43 | -2.98 | -0.61 | -0.76 |
| 183±3 GHz | 0.67 | 0.73 | 0.33 | 0.51 | -0.06 | 0.13 | -0.05 | 0.45 |
| 183±7 GHz | 1.40 | 1.27 | 0.80 | 0.80 | 1.40 | 1.27 | 0.50 | 0.56 |
| 243 GHz | 0.88 | 1.09 | 0.30 | 0.38 | 0.72 | 0.49 | 0.03 | -0.09 |
| 325±1.5 GHz | 0.88 | 0.98 | 0.62 | 0.58 | -0.13 | -0.87 | 0.31 | 0.25 |
| 325±3.5 GHz | 1.45 | 0.98 | 1.30 | 1.03 | 0.90 | 0.25 | 0.64 | 0.35 |
| 325±9.5 GHz | 0.53 | 0.82 | 0.64 | 0.74 | -0.09 | -0.73 | -0.36 | -0.74 |
| 448±1.4 GHz | 4.41 | 4.66 | 1.36 | 1.24 | -4.41 | -4.66 | -1.36 | -1.24 |
| 448±3.0 GHz | 1.50 | 1.68 | 1.43 | 1.42 | -1.28 | -1.68 | 0.41 | 0.39 |
| 448±7.2 GHz | 2.13 | 2.47 | 1.56 | 1.46 | -2.13 | -2.47 | -1.44 | -1.45 |
| 664 GHz | 1.01 | 1.00 | 1.49 | 1.50 | 0.59 | 0.68 | 1.03 | 1.44 |
| Mean | 1.31 | 1.41 | 0.84 | 0.85 | -0.42 | -0.77 | -0.12 | -0.21 |
| Mean (>=183 GHz) | 1.58 | 1.70 | 0.95 | 0.95 | -0.63 | -0.96 | -0.08 | -0.08 |

respectively. This combination will usually lead to the coldest and warmest brightness temperatures respectively, although this is not the case for channels with very low atmospheric absorption over a reflective surface. The lowest altitude sampled by the aircraft varied between flights, but was generally less than 600 m. For the aircraft in-situ profiles, data for heights below this were taken from dropsondes.

In this example there is some variability in the observed brightness temperatures along the run due to changes in the atmospheric profiles. Due to the relatively low water vapour content in this flight some channels also show variations due to changes in surface temperature and emissivity. The changes are relatively well captured by both the dropsondes and the NWP model, and there is generally good agreement between the observations and the simulations using both the dropsondes and the shortest lead-time NWP forecast. The range of simulated brightness temperatures from the estimated in-situ profiles can be quite large,

and mostly spans the range of the observations. For this flight the biggest differences between the observations and dropsonde simulations occur in the surface-sensitive channels and are likely caused by errors in the surface temperature or emissivity.

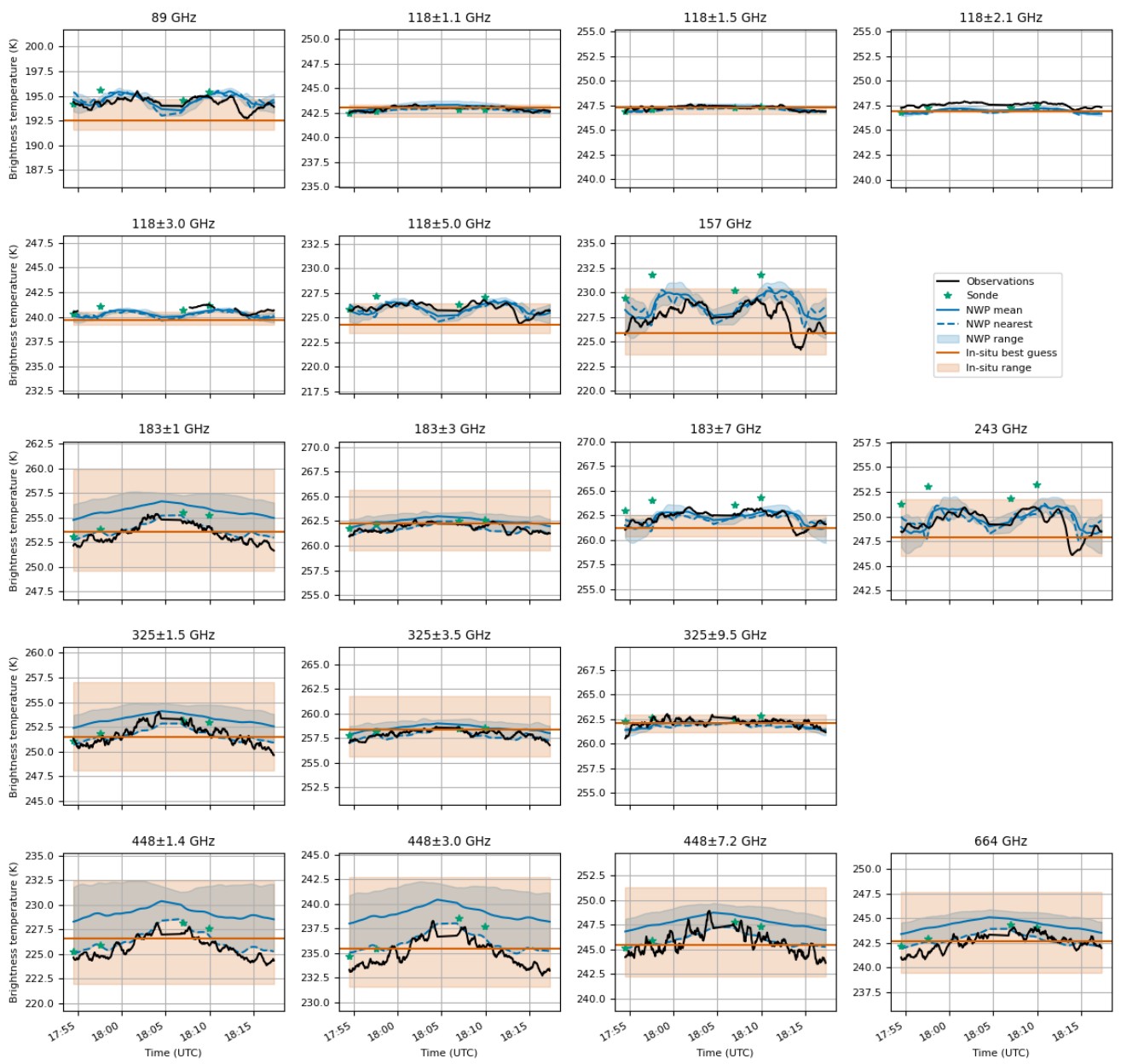

**Figure 8.** Observed and simulated nadir brightness temperatures for the high-altitude run from flight B893 using the AMSUTRAN absorption model. The black lines show the observations (smoothed over 30 s) and the stars show the simulations using the dropsonde atmospheric profiles, plotted at the time corresponding to the dropsonde release. The simulation using the best-estimate aircraft in-situ profile is shown by the orange horizontal line, with the shading indicating the range of brightness temperatures obtained using the extreme wet and dry profile estimates. The blue lines and shading show the simulations using the NWP model fields. The solid line and shading shows the mean and range of the simulations from different forecast runs, and the dashed line shows the simulation using the shortest lead-time forecast.

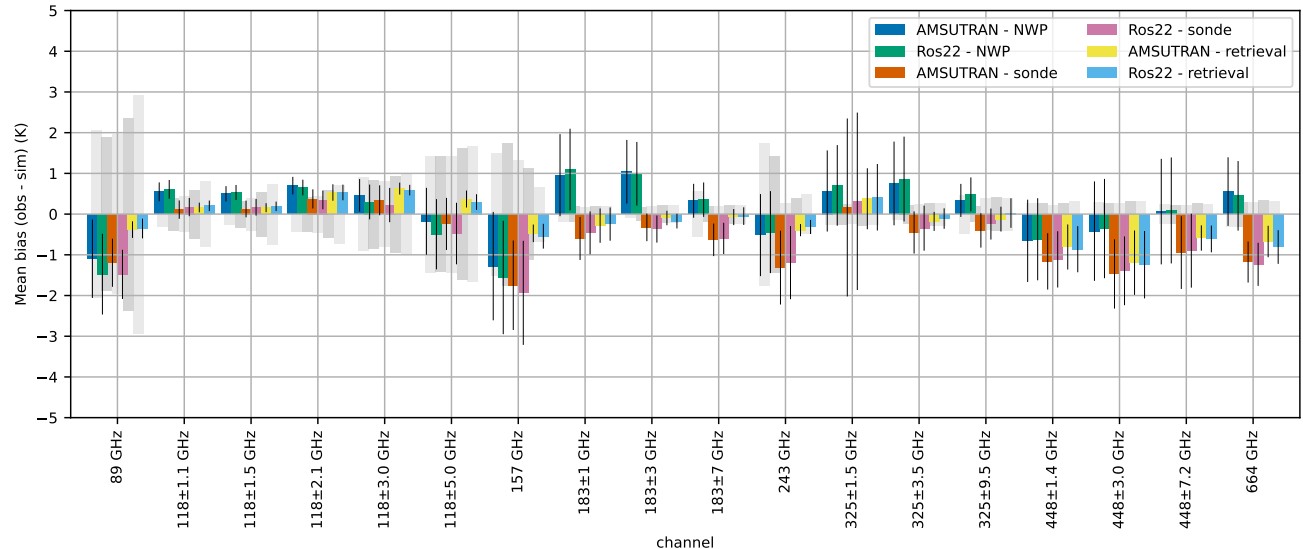

**Figure 9.** Flight-mean bias (observation - simulation) for nadir views during high-altitude runs. The black error bars show the estimated uncertainty (95% CI) in the flight-mean value. The gray shading shows the spectroscopic uncertainty estimates (95% CI) for top-of-atmosphere views at nadir incidence based on the method of Gallucci et al. (2024) for typical (from left to right) sub-arctic winter, midlatitude winter, sub-arctic summer, midlatitude summer and tropical atmospheric profiles.

The mean brightness temperature bias was calculated for each flight using both the dropsonde atmospheric profiles and the shortest lead-time NWP model forecast (which is always within 6 h of the analysis time). For the dropsonde profiles, only the brightness temperature observations made within 60 s of the sonde release time were used, corresponding to a horizontal distance of approximately 10 km. Figure 9 shows the mean bias across all flights for simulations using the AMSUTRAN and Ros22 absorption models. Also shown are the spectroscopic uncertainty estimates (95% CI) for nadir views, that have been calculated based on the method of Gallucci et al. (2024) for five atmospheric profiles ranging from sub-arctic winter to tropical conditions. The mid-latitude winter and mid-latitude summer profiles are most representative of the conditions encountered during the majority of the flights.

The difference between the AMSUTRAN and Ros22 absorption models is rather small, and is often considerably less than the estimated spectroscopic uncertainty. The difference between the absorption models is also small compared to the bias between the observations and simulations. This bias is often greater than the spectroscopic uncertainty, particularly for the channels centred on the water vapour absorption lines, as these have the smallest spectroscopic uncertainties. However, the variability of the bias between the different flights means that there is still considerable uncertainty in the flight-mean values as shown by the error bars in fig. 9, which are calculated from the standard deviation of the bias across all flights and the number of flights. This variability is driven by errors in the NWP forecast model fields, and uncertainties in the dropsonde and

radiometric measurements. A much larger dataset is needed to reduce the uncertainty in these comparisons, and this is difficult to achieve from an airborne platform.

The largest biases are found in the window channels at 89 and 157 GHz, consistent with the results of Moradi et al. (2020). These channels have the largest spectroscopic uncertainty due to the contribution from the water vapour continuum, which is also significant at 243 GHz in cold, dry conditions. However, it is also likely that they are affected by errors in the surface emissivity. Comparisons with the bias for off-nadir views (not shown), which are sensitive to surface polarisation effects, show the biggest changes in bias with viewing angle for these channels implying that the surface is having an impact. For the remaining channels the bias is less than 1.5 K for both NWP model and dropsonde simulations with both absorption models. However, there are systematic differences between the biases for the NWP model and dropsonde simulations, with the dropsondes leading to consistently warmer simulated brightness temperatures. This implies that the two sources of profile information have different characteristics, with the dropsonde profiles containing less water vapour on average. These differences suggest the atmospheric profiles are likely to be a significant contributor to the overall bias, although systematic errors in the brightness temperature observations may also be important. Fox et al. (2017) estimate that the systematic bias in the downward-looking ISMAR observations is less than 0.4 K, with the exception of 118$\pm$3 and 664 GHz where standing-wave effects may lead to errors up to 1.7 K.

Note that we have not attempted to restrict these results to homogeneous scenes as this could overly restrict a relatively small dataset. As an indication of the inhomogeneity we have calculated the standard deviation of the MARSS brightness temperatures (at 89, 157 and 183 GHz) over the 120 s window centred on the sonde release times. Averaged across all the sondes this is between 0.6 and 1.2K, with worst-case values for a single sonde between 1 and 2K, dependent on channel. The standard deviation is greatest for the 157 and 183$\pm$1 GHz channels, and smallest for the 89 and 183$\pm$7 GHz channels. Note that this figure also includes effect of the instrument NE$\Delta$T, which for the MARSS channels is typically between 0.3 and 0.65K. This suggests that the impact of inhomogeneity is generally quite small, but cannot be considered negligible. However, we would expect the impact of inhomogeneity to be reduced when averaging results across multiple sondes.

The retrieval method described in the previous section can also be applied to the down-looking observations, although it provides a less rigorous test of the absorption models than the upward-looking retrievals. In particular, the warmer radiative background leads to a lower sensitivity of the brightness temperatures to the absorption model parameters, and uncertainties in surface temperature and emissivity can influence the results. Additionally, the simultaneous use of observations from many different altitudes in the upward-looking retrieval provides a strong constraint on the vertical structure of the atmospheric profile, even for a single channel, as long as there is enough absorption to create a non-negligible vertical gradient of brightness temperature. The fit of the retrieval across multiple channels therefore provides considerable information on the accuracy of the absorption model. In contrast, for the down-looking retrieval, the information on the vertical structure can only come from using multiple channels with different weighting functions, so less information on the accuracy of the absorption model can be extracted.

Nevertheless, we have applied the retrieval to the downward-looking observations using the dropsondes as the a-priori background state. Surface temperature and near-surface wind speed were retrieved in addition to the temperature and water

vapour profiles. The background temperature error was set to 4 K and the water vapour background error was set to 20%. The background errors for the surface temperature and wind speed were set to 2 K and 3 ms$^{-1}$ respectively. The observation error was taken as the standard deviation of the brightness temperature measurements within 60 s of the sonde launch time, and is assumed to be uncorrelated between the channels. Comparing the results of the retrieval to the dropsonde simulations in fig. 9 it can be seen that the main differences occur in the window channels at 89, 157 and 243 GHz where the biases for the retrieval are reduced. This is mainly due to the retrieval reducing the near-surface wind speed which reduces the surface emission. There are also small improvements to the bias in almost all of the water vapour channels due to small adjustments made to the water vapour profile.

The target radiometric accuracy for ICI is 1 K for the 183 GHz channels and 1.5 K at higher frequencies (Eriksson et al., 2020). It is planned that these will be validated during the post-launch calibration and validation period through radiative closure calculations. The results presented in this section show that the source of atmospheric profile information can have a significant impact on these comparisons, and it is recommended that a wide range of profile data, including radiosondes, NWP models and reanalyses, are used for this purpose. However, we have shown that using both the AMSUTRAN and Ros22 absorption models it is possible to obtain mean biases close to this level of agreement for the airborne data.

## 6   Conclusions

In this study, airborne observations between 89 and 664 GHz for clear-sky scenes have been used to evaluate two models for atmospheric gas absorption using radiative closure calculations. Both upward and downward-looking views were considered.

Although upward-looking brightness temperature profiles are relatively sensitive to the absorption model, they are also strongly affected by uncertainties in the atmospheric water vapour profile and differences between observed and simulated brightness temperatures of over 20 K were seen in some cases, with large variability between different flights. The impact of uncertainties in the representativeness of the water vapour profile can be reduced by averaging across multiple flights. Alternatively, a method of retrieving the water vapour profile based on the radiometric observations has been demonstrated which leads to significantly more consistent results between the flights. Both methods gave similar results, with the exception of the two water vapour channels closest to the centres of the absorption lines at 183 and 448 GHz, which are strongly affected by the small amounts of water vapour in the upper troposphere and stratosphere which is difficult to measure accurately. For these channels the retrieval was able to provide a better fit to the observations by systematically reducing the water vapour at high altitudes. For the retrieved results, both the AMSUTRAN and Ros22 models were able to give a good fit the observations, with a flight-mean MAD across all water vapour column amounts of 0.85 K. Mean differences between observed and simulated brightness temperatures were generally close to, or within, the estimated spectroscopic uncertainty.

Radiative closure calculations for downward-looking observations were performed using atmospheric profiles measured by dropsondes released from the aircraft, and also from NWP model fields. The largest differences were seen in the window channels at 89 and 157 GHz which are influenced both by the water vapour continuum and the surface properties. For the remaining channels the flight-mean bias was less than 1.5 K for both the AMSUTRAN and Ros22 models using dropsonde

and NWP profiles. The flight-mean bias for the channels centred on the water vapour absorption lines, and also at 664 GHz was greater than the estimated spectroscopic uncertainty. Systematic differences between the two sources of profile information suggest that uncertainty in the atmospheric profile makes a significant contribution to the bias for these comparisons.

The results of this study suggest that both the AMSUTRAN and Ros22 models are sufficiently accurate for use across all the ICI channels. The AMSUTRAN model has already been used to train the fast RTTOV model that will be used operationally for ICI exploitation. Similar radiative closure calculations will also be used for calibration and validation of ICI radiometric accuracy during the post-launch commissioning phase. This is necessary because no operational satellite sensors exist at ICI frequencies above 183 GHz to enable direct instrument intercomparison. For the downward-looking results presented here we obtained flight-mean biases at frequencies above 183 GHz very close to the 1.5 K target radiometric accuracy for ICI. Given the influence of the atmospheric profile uncertainty on these comparisons it will be necessary to average across a large number of observations to obtain robust results. It is also recommended that different sources of profile information, including radiosondes, NWP models and reanalyses, are used to characterise the impact of systematic biases.

Although this study has shown good agreement between observed and simulated brightness temperatures, the results will be affected by any systematic biases in the measurements. Given the relatively small estimated spectroscopic uncertainty, the close agreement between the AMSUTRAN and Ros22 models, and the impact of the uncertainties in the atmospheric profile, it is challenging for such radiative closure calculations to discriminate between the absorption models or to show where further refinement of the spectroscopic parameters could lead to improved results. It would also be desirable to confirm the results of this study with independent measurements from different instruments.

*Code and data availability.* Observations from the FAAM aircraft are available via the CEDA archive (Facility for Airborne Atmospheric Measurements et al., 2024). The processed data used in this study are available under license for non-commercial purposes and on the condition of no redistribution by contacting EUMETSAT (vinia.mattioli@eumetsat.int), and will be made available via the project webpage in due course (https://www.eumetsat.int/study-atmospheric-absorption-models-using-ismar-data-0). Brightness temperature uncertainty due to spectroscopic parameter uncertainty can be calculated using the code described by Larosa et al. (2024). The spectroscopic parameter uncertainty covariance matrix is available as a supplement to Gallucci et al. (2024). The ARTS radiative transfer model can be obtained from https://www.radiativetransfer.org/getarts/

*Author contributions.* VM and SF designed and led the research study. AV and SF collated the airborne dataset. ET performed a review of available absorption models and provided recommendations for this study. SF and AV analysed the airborne data. DC and DG provided the analysis of the spectroscopic uncertainty. SF wrote the initial manuscript, and all authors contributed to revisions.

*Competing interests.* Domenico Cimini is a member of the editorial board of Atmospheric Measurement Techniques and the authors have also no other competing interests to declare.

*Acknowledgements.* This work was supported by the European Organization for the Exploitation of Meteorological Satellites (EUMETSAT) through the study on "atmospheric absorption models using ISMAR data" under Contract EUM/CO/20/4600002477/VM. The airborne data were obtained using the BAe-146-301 Atmospheric Research Aircraft (ARA) flown by Airtask Ltd and managed by the FAAM Airborne Laboratory, jointly operated by UKRI and the University of Leeds. The authors would like to thank the crew and personnel involved in the airborne campaigns. We acknowledge the use of imagery provided by services from NASA's Global Imagery Browse Services (GIBS), part 520 of NASA's Earth Observing System Data and Information System (EOSDIS).

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
