# Peer review of "An evaluation of atmospheric absorption models at millimetre and sub-millimetre wavelengths using airborne observations"

_EGUsphere, 2024_

## Author Response (AR1)

**Response to RC1**

We thank the referee for their supportive review. Our response to their specific comments can be found below.

**Line 100. Why not compare the complete models with a line-by-line model?** The model which we are referring to as the "AMSUTRAN model" is, effectively, a line-by-line model. As described in section 2, it now uses the same AER line database that is also the basis for the LBLRTM and MonoRTM models, along with the MT-CKD continuum. Although the "fast" version of the AER list is used, which has a reduced set of lines for computational efficiency, it still includes 338 water vapour lines (from a total of 1488 in the full AER list) and 652 ozone lines relevant to the frequency range studied. In comparison, the Ros22 model only includes 20 water vapour lines. Note that our terminology of a "complete" model [line 57] simply means that it includes both line and continuum effects, and that the line and continuum absorption models are consistent with each other. We will adjust the text to clarify our definition of a "complete absorption model" and note that the AMSUTRAN model can be considered as a line-by-line model.

New text added at line 61: *"Here, the term "complete absorption model" indicates that both line and continuum absorption are included in a consistent way."*

Modified lines 112-115 to read *"These are the updated AMSUTRAN configuration, which is effectively a line-by-line model that incorporates the AER water vapour spectroscopy, and the most recent (2022) iteration of the Rosenkranz (2017) model ... which uses a significantly reduced set of spectral lines for computational efficiency.*

**Line 209. In the case of wide antenna beams, does the use of a one-dimensional atmospheric model produce errors due to atmospheric and surface inhomogeneities, and how can this be considered?** Wide antenna beams do, of course, increase the observation footprint making it more likely that there will be inhomogeneity in the atmosphere and surface within the field of view. However, from an aircraft altitude of 10km even the widest MARSS beamwidth results in a relatively small ground footprint of approximately 2km (full-width half-maximum), and the ISMAR footprints are approximately 1km. Measurements from a downward-pointing infrared thermometer made during the lowest altitude legs of the spiral descents show that the standard deviation of the sea surface temperature over a 2km track distance is typically around 0.2K, so we expect the impact of surface inhomogeneity to be limited in these cases. It is difficult to quantify the possible impact of atmospheric inhomogeneity over the sensor footprint as there are very few reliable observations of the atmospheric state on sub-km length scales, but we do not expect it to have a significant impact on our results.

New text added at lines 234-237: *"Wide antenna beams could also introduce errors due to atmospheric and surface inhomogeneities across the antenna footprint. However, because of the low altitude of the aircraft compared to a satellite, even the widest beams have a ground footprint of approximately 2 km and we do not expect significant spatial inhomogeneities on these length-scales."*

**Line 221. Does best-estimate refer to the profile formed by the values obtained from the aircraft's in-situ measurement? What is the lowest altitude of the profile?** The "best-estimate" profiles are described in section 3 (line 176-179) and are primarily derived from the aircraft in-situ measurements. The lowest altitude in the profile varies between flights but is generally below 600m. For the "best-estimate" profiles, the

region below the minimum altitude sampled by the aircraft was taken from dropsonde measurements, although this is only relevant for the downward-looking simulations. We will clarify this in the revised text.

Line 240-241 modified to read: *"Upward-looking brightness temperatures were simulated for all run altitudes during the spiral descents for each flight using the best-estimate atmospheric profiles that were derived from the aircraft in-situ measurements as described in sec. 3."*

Added lines 392-394: *"The lowest altitude sampled by the aircraft varied between flights, but was generally less than 600 m. For the aircraft in-situ profiles, data for heights below this were taken from dropsondes."*

**Line 252. Isn't the small difference due to the fact that they are less affected by water vapor?** Yes, we agree. We were trying to make the point that because the large differences are not seen for these channels which have low sensitivity to water vapour, it is likely that the water vapour uncertainty is one of the primary reasons for the large differences seen in the other channels. However, the text in the manuscript is ambiguous and we will clarify this in the revised version.

Lines 251-253 modified to read *"Since these are the channels which are least sensitive to water vapour, this suggests that the larger differences seen in the other channels are strongly influenced by uncertainties in the water vapour profile."*

**Line 268. When using the OEM retrieval, the simulated values are undoubtedly fitted to the observed values, so the difference of the O-B values using the retrieval profiles is small. However, how to show that the retrieval profile is accurate?** Indeed, it is not surprising that the retrieval reduces the O-B difference, since that is what it is designed to do. We cannot prove that the retrieved profiles are accurate; only that we have been able to adjust our original "best estimate" in-situ profile within plausible limits, informed by the observed atmospheric variability, in order to provide a better match to the observed brightness temperatures. We have produced a figure (S4 in the supplement) comparing the a-priori background and retrieved profiles, which shows that in the majority of cases the adjustments are small and produce qualitatively plausible results. The largest changes are generally above the maximum altitude sampled by the aircraft, with retrievals from some flights showing a significant drying, and in some cases warming, compared to the background profile. We also emphasize that the fact that the retrieval is able to simultaneously improve the fit to the observations across multiple absorption lines is an indication that it is likely that errors in the atmospheric profile are the key contributor to O-B differences, because any errors in the absorption model parameters should not be strongly correlated between the different lines. We will expand on this discussion in the revised text.

Added lines 336-342: *"Although the retrieval gives a better fit to the observed brightness temperatures it is difficult to show that the retrieved profile is accurate. However, fig. S4 in the supplement compares the retrieved and a-priori background profiles, and shows that in most cases the adjustments made by the retrieval are small and within the observed atmospheric variability. The largest changes are predominantly above the maximum altitude sampled by the aircraft, where the background state is taken from an NWP model. The fact that the retrieval is able to simultaneously improve the fit to observations across multiple absorption lines is also an indication that errors in the atmospheric profile are a main cause of the difference between observation and simulation, because any errors in the absorption model parameters should not be strongly correlated between the different lines."*

**Response to RC2**

We thank the referee for their positive review. Our response to their specific comments can be found below.

**Page 6 line 149 : What is meant by vertical profiles of upward looking brightness temperatures? Do you refer to different observation angles and thus sensitivity to different altitudes? This sentence is a little bit unclear and needs some clarification.** By a "vertical profile of upward-looking brightness temperatures" we refer to the zenith brightness temperature measured at many different heights throughout the atmosphere. In this study these are measured by using the stepped spiral descent flight manoeuvre described in section 3 (line 153-159). We will clarify this in the revised text.

Lines 160-162 modified to read *"As discussed by Hewison (2006), vertical profiles of upward-looking brightness temperatures (i.e. measurements of zenith brightness temperatures made from many different altitudes) have a strong sensitivity to the absorption model..."*

**Page 10, Line 205: What is does top-hat response mean? I am not familiar with that term, so I am wondering.** By "top-hat response" we mean a uniform flat spectral response function over the channel passband. We will change the text to remove the unclear terminology.

References to *"top-hat response"* have been replaced with *"uniform spectral response function"*.

**Page 11, line 245: You refer to the standing wave effects that affect the results of the comparison. How do they look like? Is it correct that these are systematic but varying from flight to flight?** The standing wave effect is described in Sec 4.3 of Fox et al. 2017 and affects the 118±3 GHz and 664 GHz channels. We have observed that the apparent brightness temperatures of the on-board calibration targets are sensitive to small (less than 1°) changes in viewing angle, with worst-case differences of up to ±1K for 664-V, and attribute this to standing wave effects due to coherent backscatter from the targets. This will introduce a systematic bias in the measurements, but is likely to be sensitive to small changes in instrument geometry that might be caused e.g. by thermal expansion as the instrument is exposed to different environmental conditions, so can change over the course of a single flight.

Lines 288-289 modified to read *"...with some channels also impacted by standing wave effects (118±3 and 664 GHz) which can cause an additional slowly-varying systematic error."*

**Page 11 Line 232 ff: You refer to the spectroscopic uncertainties at several places. What do these uncertainties refer to? It would be especially interesting for the window channels that show quite large error bars related to the spectroscopic uncertainties.** The spectroscopic uncertainties refer to the uncertainty in the brightness temperature simulations due to the uncertainty in the spectroscopic parameters within the gas absorption models, such as the continuum strength, line strength, line width, temperature dependence etc. This is discussed in detail in the companion paper by Gallucci et al. (2024). In the window channels the dominant source of the uncertainty is the water vapour continuum and oxygen line mixing parameters. In the revised text we will discuss the Gallucci et al. study more prominently.

New text added (lines 98-102): *"When performing radiative closure experiments it is important to consider the impact of uncertainties in the spectroscopic parameters on the simulated brightness temperatures. Cimini et al. (2018) showed how this could be applied to ground-based microwave radiometers, and a recent study by Gallucci et al. (2024), performed as part of the same EUMETSAT project, has extended the analysis to sub-millimetre frequencies and satellite and airborne viewing geometries."*

Added lines 270-273: *"This is the uncertainty in simulated brightness temperatures caused by imperfect knowledge of the spectroscopic parameters in the absorption model, and here we show a 2-σ uncertainty, equivalent to the 95% CI. The largest uncertainties are seen in the window channels, where the dominant*

*source is the water vapour continuum. At 89 GHz the oxygen line mixing parameters also have a significant impact."*

**Page 12 line 267: I like the approach to retrieve profiles and then simulations based on these retrieved profiles. and apparently this improves the result, which is not really surprising the retrieval is based o the observed radiometric measurements. I am wondering, how realistic the retrieved profiles are. From the text, I get the impression that H2O and T are retrieved simultaneously. Is this the case? As the H2O retrieval is depending on the T retrieval and, e.g. for water lines the Temperature profile is affected by the H2O, it would be interesting to see if the retrieved profiles are reasonable.**

Please see our response to RC1 regarding whether the retrieved profiles are reasonable. Water vapour and temperature are simultaneously retrieved.

Lines 307-308 modified to read *"... the optimal estimation method (OEM) was used to simultaneously adjust the temperature and water vapour profiles ..."*

**Are the profile retrievals based on single measurements or on averaged measurements? Also, related to the question above, what do the brightness temperatures observed at all altitudes mean? Does this refer to the flight altitude?** "All altitudes" applies to the altitude of each flight level of the stepped spiral descent that is used to measure the vertical profile of brightness temperature. The brightness temperatures are averaged over each level segment to produce the vertical profile that is input to the retrieval. We have added a new diagram (figure 5) giving an overview of the inputs and outputs of the retrieval algorithm.

Modified lines 310-312 to read *"In the retrievals presented here, the mean brightness temperatures observed at each of the altitudes sampled by level runs during the stepped spiral descent are used simultaneously to retrieve a single atmospheric state."*

**Page 17 Fig 6: I think in the caption could be noted that this is for the upward looking cases. Page 19 tab 3: As for Fig 6, you could mention that this refers to the upward looking geometry.**

Captions adjusted on fig. 4, fig. 7 (was fig. 6) and tab. 4 (was tab. 3) to refer to upward-looking brightness temperatures.

**About section 5.2: Did you consider to apply the retrieval approach even for the down-looking observation?** The retrievals can be applied to the down-looking observations, but for several reasons they will provide a less rigorous test of the absorption models. Uncertainties due to surface properties, and the warmer radiative background in this geometry, leads to a lower sensitivity of the brightness temperatures to the absorption model parameters. Additionally, for the upward-looking observations we used the vertical profile of observed brightness temperatures at many different altitudes as input to the retrieval. This provides quite a strong constraint on the atmospheric profile, since, for a given spectroscopic model, a single channel that has sufficient absorption to create a non-negligible vertical gradient of brightness temperature but is not fully saturated within a short distance from the aircraft, is sufficient (if the temperature profile is known) to constrain the vertical profile of water vapour. The fit of the retrieval across multiple channels can therefore provide considerable information on the suitability of the absorption model. In contrast, for the downward-looking observations, multiple channels are needed to determine the vertical variability of the atmospheric state, so less information on the accuracy of the absorption model can be extracted. Nevertheless, we have now applied the retrieval to the downward-looking observations and will include the results and associated discussion in the revised text. The biggest differences occur in the window channels at 89, 157 and 243 GHz where the bias is significantly reduced. This is mainly due to the retrieval reducing the near-surface wind

speed which reduces the surface emission. There is also small improvements to the bias in almost all of the water vapour channels due to adjustments made to the water vapour profile. Fig. 9 in the revised paper (was fig. 8) now includes results from retrievals applied to down-looking observations.

New text added (lines 439-458): *"The retrieval method described in the previous section can also be applied to the down-looking observations, although it provides a less rigorous test of the absorption models than the upward-looking retrievals. In particular, the warmer radiative background leads to a lower sensitivity of the brightness temperatures to the absorption model parameters, and uncertainties in surface temperature and emissivity can influence the results. Additionally, the simultaneous use of observations from many different altitudes in the upward-looking retrieval provides a strong constraint on the vertical structure of the atmospheric profile, even for a single channel, as long as there is enough absorption to create a non-negligible vertical gradient of brightness temperature. The fit of the retrieval across multiple channels therefore provides considerable information on the accuracy of the absorption model. In contrast, for the down-looking retrieval, the information on the vertical structure can only come from using multiple channels with different weighting functions, so less information on the accuracy of the absorption model can be extracted.*

*Nevertheless, we have applied the retrieval to the downward-looking observations using the dropsondes as the a-priori background state. Surface temperature and near-surface wind speed were retrieved in addition to the temperature and water vapour profiles. The background temperature error was set to 4 K and the water vapour background error was set to 20%. The background errors for the surface temperature and wind speed were set to 2 K and 3 $ms^{-1}$ respectively. The observation error was taken as the standard deviation of the brightness temperature measurements within 60 s of the sonde launch time, and is assumed to be uncorrelated between the channels. Comparing the results of the retrieval to the dropsonde simulations in fig. 9 it can be seen that the main differences occur in the window channels at 89, 157 and 243 GHz where the biases for the retrieval are reduced. This is mainly due to the retrieval reducing the near-surface wind speed which reduces the surface emission. There are also small improvements to the bias in almost all of the water vapour channels due to small adjustments made to the water vapour profile."*

**Response to RC3**

We thank the referee for their thorough and insightful review. Our response to their comments is below.

**General comments**

**As mentioned above, the (non-negligible) discrepancies between observed and simulated TBs might have different reasons, though I agree with the authors that the most likely reason is the lack of knowledge on water vapor. In fact, the major finding of the manuscript for me is the difficulty of getting the atmospheric state right and not on the absorption models. Therefore, the discussion should be extended to make this clear, and also it should be reflected in the abstract as a call to the community to work on this highly relevant topic.** Indeed, we believe that getting the correct atmospheric state is key to the success of a radiative closure study such as this, and this is difficult to do, even with a relatively comprehensive suite of airborne atmospheric measurements. Humidity is particularly challenging, partly due to its spatial inhomogeneity, and also due to challenges accurately measuring low values of specific humidity which still have significant impact on channels close to the centre of absorption lines. Nevertheless, we believe that we have shown that it is still possible to perform a meaningful evaluation of absorption models, either by averaging across a (preferably large) number of cases to reduce the impact of errors in

specific atmospheric profiles, or by using a method such as the retrievals presented in this study to demonstrate the consistency of the absorption model across multiple channels using different spectroscopic lines. We will expand on this point in the revised text.

Abstract (line 6-9) modified to read *"Differences of 20 K are seen in some individual comparisons, with the largest discrepancies occurring where the brightness temperature is highly sensitive to the atmospheric water vapour profile. However, these differences are within the expected uncertainty due to the observed water vapour variability, highlighting the importance of understanding the spatial and temporal distribution of water vapour when performing such comparisons."*

Line 259-265 modified to read *"Water vapour can be highly variable, even over the relatively compact area sampled by the spiral descents, meaning that the best-estimate in-situ measurements may not adequately represent the profiles at the time and location of the radiometric observations. This is further demonstrated by the large spread of simulated brightness temperatures when using the atmospheric profiles based on the extremes of the in-situ observations, which is generally larger than the brightness temperature differences between observations and simulations. This highlights the importance of accurately measuring the water vapour profile across the field of view of the instrument at the time of measurement when performing such closure studies, although this is difficult to achieve in practice."*

**The title emphasizes the intercomparison of both absorption models. Thus a Table comparing the different ingredients would be very helpful, and easier to comprehend than the text I section 2. Also the work by Galluci et al on uncertainties should be mentioned here. At the end of Section 5.1. the authors conclude that both have similar performance but the interesting question is whether this is because they are similar in their settings or whether compensating effects exist? That there are differences becomes clear in Fig. 6 for the innermost channels of the 183 and 448 lines – this needs to be understood better (see my point 1)**

We have included a new table (tab. 1) summarising difference between the models in section 2. There are many differences in the details between the two models, but it is beyond the scope of this study to try and separate the individual contributions. The report by Turner et al., 2022 NWPSAF-MO-TR-039 provides a detailed comparison of several absorption models. I am unclear what the referee means regarding Figure 6. The biggest differences for the innermost channels of the 183 and 448 GHz lines shown in the figure is between the results using the "best estimate" atmospheric profiles and the retrievals. The differences between the two absorption models are mostly fairly small.

Work by Gallucci et al. now described in introduction (line 100-102): *"Cimini et al. (2018) showed how this could be applied to ground-based microwave radiometers, and a recent study by Gallucci et al. (2024), performed as part of the same EUMETSAT project, has extended the analysis to sub-millimetre frequencies and satellite and airborne viewing geometries."*

New text added discussing Turner et al., 2022 (line 107-110): *"A review and comparison of absorption models applicable across the microwave and sub-millimetre spectral range was performed by Turner et al. (2022). This includes several commonly used absorption models, as well as the configuration of AMSUTRAN used for RTTOV v12 (described by Turner et al., 2019) and an updated version that was designed to improve its validity in the sub-millimetre spectral region."*

**The comparison includes two state-of-the-art absorption models. But these are not the ones that are most used in the community today. Though AMSUTRAN is going to be the basis for the next version of RTTOVS, the more important question is how the current version behaves (at least for 89 and 183**

**GHz channels)? Or the ones that have been used for the generation of ERA5 or other frequently used models (Liebe93, etc)? I completely understand if the authors don't want to include this in full detail but some rough estimation and general statement would be important for many readers.** A main motivation for this study was to explore the applicability of the absorption models to the sub-millimetre region that will be observed from space by ICI. Hence we have focussed on models which have been developed to cover this spectral range, and the AMSUTRAN version that has used to produce the RTTOV coefficients for ICI. The report by Turner et al., 2022, referenced above, includes a comparison of many absorption models (see, for example, Fig 23). We have included a brief discussion of this report in the revised paper.

**The paper is relatively poor with respect to "open science" issues and the reproducibility of the results. Maybe the authors can reconsider data publication of drop sondes and radiometer measurements. As no scripts are made available some details on data processing might be given in an appendix. E.g. hardly any information is available on the retrieval scheme.** The original level-1 airborne data are available via the FAAM archive on CEDA (https://data.ceda.ac.uk/badc/faam/data), but further data processing and quality control was applied to produce the dataset used in this study. It is our intention to make the full observational dataset available via an associated project page on the EUMETSAT website (https://www.eumetsat.int/study-atmospheric-absorption-models-using-ismar-data-0) but this is yet to be put in place. In the meantime, as noted in the "code and data availability" section, the dataset is available on request from EUMETSAT. We believe this will be the simplest way for interested parties to obtain the data. The retrieval is implemented directly using the OEM capability in ARTS, available from https://www.radiativetransfer.org/getarts/

The "Code and data availability" section has been updated to read *"Observations from the FAAM aircraft are available via the CEDA archive (Facility for Airborne Atmospheric Measurements et al., 2024). The processed data used in this study are available under license for non-commercial purposes and on the condition of no redistribution by contacting EUMETSAT (vinia.mattioli@eumetsat.int), and will be made available via the project webpage in due course (https://www.eumetsat.int/study-atmospheric-absorption-models-using-ismar-data-0). Brightness temperature uncertainty due to spectroscopic parameter uncertainty can be calculated using the code described by Larosa et al. (2023). The spectroscopic parameter uncertainty covariance matrix is available as a supplement to Gallucci et al. (2024). The ARTS radiative transfer model can be obtained from https://www.radiativetransfer.org/getarts/"*

**I am rather irritated by the explicit figure captions in the text and the missing descriptions in the figure caption. I typically read the text without looking at the figures, as they should only be the proof for the statements in the text. Such detailed descriptions of different lines strongly disturb the flow of the paper. I didn't check AMT but most journals give guidelines to avoid this and also say that the captions should be explicit enough that the reader can understand the significance of the illustration without reference to the text.** We apologise for the irritation, and will reword the revised text to remove detailed figure descriptions from the main body of the text.

Section 5 has been re-ordered to remove the explicit figure information from the text.

The caption for fig. 8 (was fig. 7) has been re-written to avoid reference to the text.

**Specific comments**

**Abstract: l4-6: The text can be misinterpreted – suggest "In this study, airborne observations of clear-sky scenes between 89 and 664 GHz are used to evaluate two state-of-the-art absorption models, both**

**integrated into the Atmospheric Radiative Transfer Simulator (ARTS). Radiative closure calculations for both upward and downward-looking viewing directions…"**

Abstract (line 3-6) modified to read *"In this study, airborne observations of clear-sky scenes between 89 and 664 GHz are used to perform radiative closure calculations for both upward and downward-looking viewing directions in order to evaluate two state-of-the-art absorption models, both integrated into the Atmospheric Radiative Transfer Simulator (ARTS)."*

**Introduction: The "sister" paper by Gallucci et al. (2023) is quite important for understanding but not introduced in the beginning.**

As noted above, this paper is now discussed in the introduction (line 100-102).

**P2, l29 "The brightness temperatures for cloudy scenes are generally reduced compared to the equivalent clear-sky value due to scattering from the ice crystals in the cloud" – make clear that this holds for (thick) ice clouds only**

Line 30-32 modified to read *"For thick ice clouds, which are dominated by scattering, the brightness temperatures for cloudy scenes are generally reduced compared to the equivalent clear-sky value ..."*

**P2l95: Say that they are integrated in ARTS. What is the uncertainty of using this model?** It is difficult to assign a specific uncertainty to a particular radiative transfer model. Aside from the uncertainty of the gas absorption, the main uncertainty from the radiative transfer model (assuming known atmospheric profiles) comes from the method used to discretise the radiative transfer equation, and the assumptions within the model on how the atmospheric profiles vary between the discrete vertical levels provided as inputs. These differences generally reduce as the vertical resolution is increased. Melsheimer et al. (2005) compared the version of ARTS available at the time to three other models for downward-looking AMSU-B simulations and found that, when consistent spectroscopy was implemented across the models, differences were less than ~1K with the exception of one outlying model at 183 GHz. Much smaller differences (~0.1K) were seen for upward-looking simulations. We have also compared ARTS with AMSUTRAN at frequencies up to 1 THz using consistent spectroscopy, and found that, with sufficiently high vertical resolution, the differences in down-looking brightness temperatures were generally less than 0.1K except for very close to the centre of some absorption lines.

Line 96-98 modified to read *"These data are used to perform radiative closure calculations with the two gas absorption models, both of which have been integrated into the Atmospheric Radiative Transfer Simulator (ARTS, Buehler et al., 2024) to determine their performance at the frequencies of interest."*

New text added (line 296-303): *"An additional uncertainty also arises from the choice of radiative transfer model. Even with consistent spectroscopy, different models do not produce identical results, mainly due to the method used to discretise the radiative transfer equation and the assumptions within the model on how the atmospheric profiles vary between the discrete vertical levels provided as inputs. The differences between models generally reduces as the vertical resolution is increased. Melsheimer et al. (2005) compared the version of ARTS available at the time to three other models for downward-looking AMSU-B simulations and found that, when consistent spectroscopy was implemented across the models, differences were less than ~1K with the exception of one outlying model at 183 GHz. Much smaller differences (~0.1K) were seen for upward-looking simulations. We have tried to minimise the uncertainty due to the radiative transfer model in this study by performing calculations with a high vertical resolution."*

**P6l163: identifying regions of enhanced brightness temperature is very vague – I understand you want to be conservative** Cloud-screening based on the 89 GHz brightness temperature was only applied to two cases (B940 and C244) where visual observations during the flight identified the presence of small amounts of low-level cloud. The process was partly subjective, but observations which had not previously been removed by lidar screening (noting that the lidar was not fitted during C244) where the 89 GHz brightness temperature was enhanced by approximately 2K compared to the rest of the run were removed. Only a very small number of observations were removed in this way, representing approximately 90 seconds of data.

Line 174-176 modified to read "*A small number of additional observations were also removed, where visual observations noted the presence of low cloud in the area and the brightness temperatures at 89 GHz were enhanced by ~2 K compared to the rest of the run.*"

**P7/P8 Dropsonde description is short. Do you use the ASPEN software for corrections. Do you calculate variability in atmospheric to filter for homogeneous scenes? What is the impact of the 15 % humidity correction.** All dropsondes were processed using ASPEN v3.3 (minor revision 265-543 dependent on flight). The TDDryBiasCorr processing option was applied for the RD94 sondes. The 15% humidity correction for the RD94 sondes was derived from a separate intercomparison flight in March 2022 where three sets of sondes, consisting of both RD94 and RD41 sonde types, were released in rapid succession (Hannah Price, FAAM, personal communication). The comparison showed a consistent bias of around 15% in the humidity for the RD94s compared to the RD41s. Note that the RD94s used in this study were not reconditioned to remove contamination from the humidity sensor because the option was not available with the AVAPS system on the aircraft at the time; the March 2022 flight demonstrated that the reconditioning process significantly improved the agreement between the RD94 and RD41. The 15% humidity correction improved the fit between the observed and simulated brightness temperatures when using the dropsonde profiles, and it also improved the consistency of the results between the flights using the different sonde types. There was a single flight (C248) where both sonde types were used, and the 15% correction gave a closer match between the simulated brightness temperatures for the two sonde types. For these reasons we believe that the correction is plausible.

We have not attempted to filter for homogeneous scenes as this would overly restrict a relatively small dataset. As an indication of the inhomogeneity we have calculated the standard deviation of the MARSS brightness temperatures over the 120 second window centred on the sonde release time. Averaged across all the sondes this is between 0.6 and 1.2K, with worst-case values for a single sonde between 1 and 2K, dependent on channel. The standard deviation is greatest for the 157 and 183±1 GHz channels, and smallest for the 89 and 183±7 GHz channels. Note that this figure also includes effect of the instrument NE∆T. This suggests that the impact of inhomogeneity is generally quite small, but cannot be considered negligible. However, we would expect the impact of inhomogeneity to be reduced by averaging results across multiple sondes.

Added line 195-196 "*All sondes were processed using the ASPEN software.*" We include the rest of the information above for completeness in the discussion record.

New text lines 431-438: "*Note that we have not attempted to restrict these results to homogeneous scenes as this could overly restrict a relatively small dataset. As an indication of the inhomogeneity we have calculated the standard deviation of the MARSS brightness temperatures (at 89, 157 and 183 GHz) over the 120 s window centred on the sonde release times. Averaged across all the sondes this is between 0.6 and 1.2K, with worst-case values for a single sonde between 1 and 2K, dependent on channel. The standard deviation*

*is greatest for the 157 and 183±1 GHz channels, and smallest for the 89 and 183±7 GHz channels. Note that this figure also includes effect of the instrument NE∆T, which for the MARSS channels is typically between 0.3 and 0.65K. This suggests that the impact of inhomogeneity is generally quite small, but cannot be considered negligible. However, we would expect the impact of inhomogeneity to be reduced when averaging results across multiple sondes."*

**L215 delete blank before .** Thank you for pointing this out, we will correct it in the revised manuscript.

**Section 4: I would have preferred the description of the setup for the RT – construction of the atmospheric state in this section instead of lines 220 (5.1) onwards as this holds for both up and downward looking geometry.**

The atmospheric profile information is now included in sec. 4.

**Lines 227 to 238: Figure caption and different lines should not be described in the text, but the content of the plot. The same comment hold for several figures, which I don't mention later on**

As noted above, we have reworded this in the revised manuscript.

**L250: 20 K are very large – here, you need to show how much change in water vapor can explain this discrepancy (and whether this is reasonable) before just drawing the conclusion. My gut feeling is that this is the case but there needs to be more detail….especially as you say later (l275) "..that the atmosphere is sufficiently homogenous** Indeed, 20K is a large difference, but we show in figure 4 that this is within the range of uncertainty given by our extreme "wet" and "dry" profiles that were based on the range of in-situ measurements. We will include plots of the individual profiles, including the wet and dry extremes, in an appendix or as supplementary material in the revision.

New figure S1 included in supplementary material showing observed atmospheric profiles and variability. Added text line 192-193 *"The individual profiles are plotted in fig. S1 in the supplement."*

**L357: Why forecast and not (re)analysis?** We consider that the short-range forecast are likely to give a better indication of the conditions at the time of the observation than the analysis as they can account for temporal changes between the analysis and observation times. For flights B893-C158 the forecasts were run from an analysis produced on a 6-hourly cycle. For the later flights the operational model had been upgraded to run on an hourly cycle so there will be little difference between the forecast and analysis. We wanted to use a high spatial resolution product to try and capture any inhomogeneity in the atmospheric profiles, and considered that the ~31km resolution of ERA5 was insufficient.

Lines 402-403 modified to read *"The mean brightness temperature bias was calculated for each flight using both the dropsonde atmospheric profiles and the shortest lead-time NWP model forecast (which is always within 6 h of the analysis time)."*

**L390: Also reanalysis – in the conclusion you could also discuss specific experiment settings**

References to reanalyses added in lines 463 and 495.

**Section 5.2: I am missing information on the surface dependence. Do you get better agreement for calmer sea surfaces?** Unfortunately there is not enough data to determine whether there is a correlation between the brightness temperature bias and the surface wind speed.

**Figure 2: Horizontal scale needed**

Horizontal scale added to fig. 2.

**Figure 4: What does "partial column" mean – is it just the flight altitude** We mean the integrated water vapour from the flight altitude up to the top of the atmosphere as described in the text on lines 224-225.

We have clarified this in the caption to fig. 4.

**Fig. 7: Why do you show nadir if (as mentioned in the text and also shown in Fig.8 the interest is on 53 deg?** Both Fig.7 and Fig.8 show nadir results. This was chosen for simplicity as it removes the need to account for the surface-induced polarisation effects that strongly affect the window channels, and can be sensitive to small changes in aircraft pitch and roll. However, the spectroscopic uncertainty calculations of Gallucci et al. were performed for the 53° incidence angle as noted in the caption of Fig.8 in the original text. Although this introduces an inconsistency we do not expect there to be large differences in the spectroscopic uncertainty between the different incidence angles. However, to be fully consistent, the spectroscopic uncertainty estimates have now been recalculated for nadir views following the method of Gallucci et al. (2024), so the results presented in fig. 9 (was fig. 8) are now consistent with the nadir measurements.

Line 406 has been reworded to read *"Also shown are the spectroscopic uncertainty estimates (95% CI) for nadir views, that have been calculated based on the method of Gallucci et al. (2024) for five atmospheric profiles ..."* and the caption for fig. 9 (was fig. 8) has been altered in a similar way.

**Table 3. Why not show biases?** We believe that the mean absolute deviation calculated across the different partial column IWV bins, as shown in the table, is a more meaningful metric for the fit between the observations and simulations than the bias, because it is not impacted by compensating errors. However, we have also included the bias for completeness in the revised text.

Table 4 (was table 3) now includes both MAD and bias. New text added lines 363-364: *"Since the bias can be small in cases where there are compensating positive and negative errors at different values of IWV, the MAD is our preferred statistic for overall comparison between the models."*

**Supplement: I am missing figure captions – or at least an obvious note of their difference.** The figures in the supplement are simply larger versions of the individual panels of figures 4 and 5, so the captions are the same.

Captions have now been added to all figures in the supplement.

**Supplement Fig.4: At first glance, the figure might be misleading as the large shaded range at high water vapor points at large differences. I suggest using the second axis and the lower (empty) space to include the number of measurements per water vapor bin to (hopefully show that the majority of measurements have low differences.** Fig 4. in the supplement is simply a magnified version of the individual panels of figure 4 in the main paper. As noted in the caption of that figure, the shaded ranges represents the range of simulated brightness temperatures using plausible extreme warm/wet and cold/dry atmospheric profiles for each individual flight, based on the range of the in-situ observations. The large shaded range is an indication of the very high sensitivity of the simulated brightness temperatures to the amount of atmospheric water vapour; for channels close to the centre of absorption lines the biggest differences occur at low partial column IWV (high altitudes), and in atmospheric window regions they occur at and high partial column IWV (low altitudes).